

# Subgrid Variations of the Cloud Water and Droplet Number Concentration Over Tropical Ocean:

# Satellite Observations and Implications for Warm Rain Simulation in Climate Models

Zhibo Zhang[1,2*], Hua Song[2], Po-Lun Ma[3], Vincent E. Larson[4], Minghuai Wang[5], Xiquan Dong[6], Jianwu Wang[7]

1.  Physics Department, University of Maryland Baltimore County (UMBC), Baltimore, MD, USA

2.  Joint Center for Earth Systems Technology, UMBC, Baltimore, MD, USA

3.  Atmospheric Sciences and Global Change Division, Pacific Northwest National Laboratory, Richland, WA, USA

4.  Department of Mathematical Sciences, University of Wisconsin—Milwaukee, Milwaukee, WI, USA

5.  Institute for Climate and Global Change Research & School of Atmospheric Sciences, Nanjing University, Nanjing, China

6.  Department of Hydrology & Atmospheric Sciences, University of Arizona, Tucson, AZ, USA

7.  Department of Information System, UMBC, Baltimore, MD, USA

*Corresponding Author:
    Dr. Zhibo Zhang
    Email: Zhibo.Zhang@umbc.edu
    Phone: +1 (410) 455 6315



**Abstract**:

One of the difficulties of simulating the warm rain process in global climate models (GCM)

is how to account for the impact of subgrid variations of cloud properties, such as cloud water
and cloud droplet number concertation, on the nonlinear precipitation processes such as
autoconversion. In practice, this impact is often treated by adding a so-called enhancement
factor term to the parameterization scheme. In this study, we derive the subgrid variations of
liquid-phase cloud properties over the tropical ocean using the satellite remote sensing products
from MODIS (Moderate Resolution Imaging Spectroradiometer) and investigate the
corresponding enhancement factors for the GCM parameterization of autoconversion rate. The
wide spatial coverage of the MODIS product enables us to depict a detailed quantitative picture
of the enhancement factor $E_q$ due to the subgrid variation of cloud water, which shows a clear
cloud regime dependence, namely a significant increase from the stratocumulus (Sc) to cumulus
(Cu) cloud regions. Assuming a constant $E_q = 3.2$ would overestimate the observed $E_q$ in the Sc
regions and underestimate it in the Cu regions. We also found that the $E_q$ based on the
Lognormal PDF assumption performs slightly better than that based on the Gamma PDF
assumption. A simple parameterization scheme is provided to relate the $E_q$ to the grid-mean
liquid cloud fraction, which can be readily used in GCMs. For the first time, the enhancement
factor $E_N$ due to the subgrid variation of CDNC is derived from satellite observation, and results
reveal several regions downwind of biomass burning aerosols (e.g., Gulf of Guinea, East Coast of
South Africa), air pollution (i.e., Eastern China Sea), and active volcanos (e.g., Kilauea Hawaii and
Ambae Vanuatu), where the $E_N$ is comparable, or even larger than $E_q$, even after the optically
thin clouds are screened out.






## 1. Introduction

Clouds are a strong modulator of Earth's radiative energy budget (Klein and Hartmann, 1993; Trenberth et al., 2009). They can interact with other components of the climate system, such as ocean, land, and aerosols, in various ways. The feedback of clouds to climate change remains one of the largest uncertainties in our understanding of the climate sensitivity (Bony and Dufresne, 2005; Soden and Held, 2006). Despite their importance in the climate system, simulating clouds in conventional general circulations models (GCM) has proved to be extremely challenging. A main difficulty is rooted in the fact the typical grid size of GCM (~100km) is much larger than the spatial scale of many cloud microphysical processes, and as a result these subgrid scale processes, as well as the subgrid cloud variations, have to be highly simplified and then parameterized as functions of resolved, grid-level variables.

Of particular interest in this study is the warm rain processes in liquid-phase clouds, which have fundamental impacts on the cloud water budget and lifetime. Although in reality it is highly complicated and involves multiple factors, warm rain formation in GCMs is usually parameterized as simple functions of only key cloud parameters. For example, the drizzle in MBL cloud is initialized by the so-called autoconversion process in which the collision-coalescence of cloud droplets gives birth to large drizzle drops (Pruppacher and Klett, 1997). In GCMs, for the sake of efficiency, this process is usually parameterized as a function of liquid water content (LWC or symbol $q_c$) and cloud droplet number concentration (CDNC or symbol $N_c$) (Khairoutdinov and Kogan, 2000) (see section 2 for details). Even though this is highly simplified, the parametrization scheme still faces a great difficulty. The calculation of grid-mean autoconversion efficiency requires the knowledge of subgrid distributions of LWC and CDNC, but in the GCMs only grid-mean quantities $\langle q_c \rangle$ and $\langle N_c \rangle$ are known and available for use in the computation of autoconversion rate. As pointed out by Pincus and Klein (2000), for a process $f(x)$ such as autoconversion that is nonlinearly dependent on subgrid variables, $x$, the grid-mean value $\langle f(x) \rangle$ is not equal to the value estimated based on the grid-mean $\langle x \rangle$, i.e., $\langle f(x) \rangle \neq f(\langle x \rangle)$. Mathematically, if $f(x)$ is convex, then $f(\langle x \rangle) < \langle f(x) \rangle$ (Larson and Griffin, 2013; Larson et al., 2001). To take this effect into account, a parameter $E$ is often introduced in the GCM as part of the parameterization such that $\langle f(x) \rangle = E \cdot f(\langle x \rangle)$. It is referred to as the "enhancement factor"



in many studies and this study too because $E > 1$ for a convex function. Such a nonlinear effect
is not just limited to the autoconversion process. Some other examples are the plane-parallel
albedo bias (Barker, 1996; Cahalan et al., 1994; Oreopoulos and Davies, 1998a), subgrid cloud
droplet activation (Morales and Nenes, 2010) and accretion (Boutle and Abel, 2012; Lebsock et
al., 2013).

The value of $E$ is determined primarily by two factors: the nonlinearity of $f(x)$ and the

subgrid probability density function (PDF) $P(x)$. Given the same subgrid variation of LWC, i.e.,
$P(q_c)$, the nonlinear effect impacts the autoconversion process more than it does on the
accretion process, because the former is a more nonlinear function of $q_c$ than the latter. For the
same $f(x)$, a grid box with a narrow and symmetric $P(x)$ would require a smaller $E$ than another
grid box with a broader and non-symmetric $P(x)$. The shape of the $P(x)$ is dependent on mainly
on cloud regime. Take cloud water for example. The subgrid PDF of cloud water $P(q_c)$ is usually
narrower and more Gaussian-like in the stratocumulus (Sc) region while in the broken cumulus
(Cu) cloud region, $P(q_c)$ is usually broader and more skewed (Barker et al., 1996; Lee et al., 2010;
Oreopoulos and Cahalan, 2005; Wood and Hartmann, 2006). Obviously, model resolution is also
an important factor—the coarser the spatial resolution, the larger the subgrid cloud
inhomogeneity. Ideally, the value of the enhancement factor $E$ should be diagnosed from the
subgrid cloud PDF $P(x)$, which should be scale aware and dependent on cloud regime.
Unfortunately, because this is not possible in most conventional GCMs, the value of $E$ is usually
assumed to be a constant for the lack of better options. The $E$ for autoconversion due to subgrid
LWC variation is assumed to be 3.2 in the two-moment cloud microphysics parameterization
schemes by Morrison and Gettelman (2008) (MG scheme hereafter), which is employed in the
widely used Community Atmosphere Model (CAM). This choice of $E = 3.2$ is based on an early
study by Barker et al. (1996), in which the mesoscale variation of column-integrated optical
thickness of the "overcast stratocumulus", "broken stratocumulus" and "scattered
stratocumulus" are studied. The value $E = 3.2$ is derived based on the mesoscale variation of the
broken stratocumulus.

Clearly, a simple constant $E$ is not adequate. The following is a list of attempts to better

understand the subgrid cloud variations and the implications for warm rain simulations in GCMs.



Several previous studies have shown that the mesoscale cloud water variation is a strong function
of cloud regime—the subgrid cloud water variation of Sc cloud is much different from that of Cu
clouds (Barker et al., 1996; Lee et al., 2010; Oreopoulos and Cahalan, 2005; Wood and Hartmann,
2006). As the first part of a two-part study, Larson and Griffin (2013) first laid out a systematic
theoretical basis for understanding the effects of subgrid cloud property variations on simulating
various nonlinear processes in GCM, including not only the autoconversion but also the accretion,
condensation, evaporation and sedimentation processes. In the second part, using cloud fields
from a large-eddy simulation (LES), Griffin and Larson (2013) showed that inclusion of the
enhancement factor indeed leads to more rainwater at surface in single-column simulations and
makes them agree better with high-resolution large-eddy simulations. Using a combination of in
situ measurement and satellite remote sensing data, Boutle et al. (2014) analyzed the spatial
variation of cloud and rain water, as well as their covariation. They further developed a simple
parameterization scheme to relate the subgrid cloud water variance to the grid-mean cloud
fraction. Recently, using the ground-based observations from three Department of Energy (DOE)
Atmospheric Radiation Measurement (ARM) sites, Xie and Zhang (2015) developed a scale-aware
parameterization scheme for GCMs to account for subgrid cloud water variation. Although these
previous studies have shed important light on subgrid cloud variation and the implications for
GCM, they lack a global perspective because they are only based on limited data (e.g., LES cases,
in situ and ground-based measurement). Currently, satellite remote sensing observation is the
only way to achieve a global perspective, although remote sensing products suffer from inherent
retrieval uncertainties. Using the observations from the space-borne radar CloudSat, Lebsock et
al. (2013) showed that the subgrid cloud water variance is larger over the Sc region than over the
Cu region, and as a result the enhancement factor shows an increasing trend from Sc to Cu region.
They also highlighted importance of considering the subgrid co-variability of cloud water and rain
water in the computation of the accretion rate.  On the modeling side, Guo et al. (2014)
investigated the sensitivity of cloud simulation in the Geophysical Fluid Dynamics Laboratory
(GFDL) Atmospheric General Circulation Model (AM) to the subgrid cloud water parameterization
schemes. A similar study  was carried out by Bogenschutz et al. (2013) using the National Center
of Atmospheric Research (NCAR) Community Atmospheric Model (CAM). Both studies show that



the more sophisticated subgrid parameterization scheme— Cloud Layers Unified by Binormals
(CLUBB) (Golaz et al., 2002a; 2002b; Larson et al., 2002)—lead to a better simulation of clouds in
the model. However, a more recent study by Song et al. (2017) reveals that the CLUBB in CAM
version 5.3 (CAM5.3) overestimates the enhancement factor in the trade wind cumulus cloud
region, which in turn leads to the "empty cloud" problem.

Despite these previous studies, many questions remain unanswered. First of all, all the

previous studies, as far as we know, have focused on the impact of subgrid cloud water variation.
The potential impact of subgrid variation of cloud microphysics, namely CDNC, has been
overlooked so far. Given the same amount of cloud water, a cloud with a smaller CDNC would
have larger droplets and therefore larger precipitation efficiency than another cloud with a larger
CDNC. Secondly, most of previous studies are based on the assumption that the subgrid cloud
property variation follows certain well-behaved distributions, usually either Gamma (e.g., Barker,
1996; Morrison and Gettelman, 2008; Oreopoulos and Barker, 1999; Oreopoulos and Cahalan,
2005) or Lognormal (Boutle et al., 2014; Larson and Griffin, 2013; e.g., Lebsock et al., 2013).
However, the validity and performance of the assumed PDF shape are seldom checked.
Furthermore, although the study by Lebsock et al. (2013) has depicted a global picture of the
enhancement factor for the autoconversion modeling in GCM, the picture is far from clear due
to the small sampling rate of CloudSat observations.

In this study, we revisit the subgrid variations of liquid-phase cloud properties over the

tropical ocean using 10 years of MODIS cloud observations, with the overarching goal to better
understand the potential impacts of subgrid cloud variations on the warm rain processes in the
conventional GCMs. Similar to previous studies, we will quantify the subgrid cloud water
variations based on MODIS observations. Going one step further, we will also attempt to unveil
for the first time the subgrid CDNC variation and investigate its implications for warm rain
simulations in GCM. Moreover, we will take advantage of the wide spatial coverage of MODIS
data to achieve a more detailed picture of the enhancement factor for the autoconversion
simulation. Last but not least, we will evaluate the two widely used distributions, i.e., Lognormal



and Gamma, in terms of their performance and limitations for simulating the enhancement
factor.

The rest of the paper is organized as follows, we will first explain the theoretical

background in Section 2 and introduce the data and methodology in Section 3. The MODIS
observations of the grid mean values and subgrid variations of key cloud properties will be
presented and discussed in Section 4. The implications for the autoconversion process simulation
in the GCMs will be discussed in 5. The main findings will be summarized in Section 6 with an
outlook for future studies.
2.  Theoretical Background

2.1. Theoretical Distributions to describe subgrid cloud property variations

In previous studies, the spatial variations of cloud properties, such as cloud optical thickness

(COT), cloud liquid water path (LWP) and cloud liquid water content (LWC), are often described
using either of two theoretical distributions—the Gamma and Lognormal distribution. The
probability density function (PDF) from a Gamma distribution is a two-parameter function as
follows (Barker, 1996; Oreopoulos and Davies, 1998b):

$$P_G(x) = \frac{1}{\Gamma(v)} \alpha^v x^{v-1} \exp(-\alpha x),  \tag{1}$$

where $\Gamma$ is the Gamma function, $v$ is the so-called inverse relative variance, and $\alpha$ the so-called
rate parameter. The mean value of a Gamma distribution Is given by

$$\langle x \rangle = \int_0^\infty x\, P_G(x) dx = \frac{v}{\alpha},  \tag{2}$$

and the variance given by

$$Var(x) = \int_0^\infty (x - \langle x \rangle)^2\, P_G(x) dx = \frac{v}{\alpha^2}.  \tag{3}$$

It follows from Eq. *(2)* and *(3)* that the inverse relative variance

$$v = \frac{1}{\eta} = \frac{\langle x \rangle^2}{Var(x)},  \tag{4}$$

where $\eta = \frac{Var(x)}{\langle x \rangle^2}$ is the relative variance.



The PDF of a Lognormal distribution is given as follows (Larson and Griffin, 2013;
Lebsock et al., 2013):

$$P_L(x) = \frac{1}{\sqrt{2\pi}x\sigma}\exp\left(-\frac{(\ln x - \mu)^2}{2\sigma^2}\right), \qquad (5)$$

where $\mu = \langle \ln x \rangle$ and $\sigma^2 = Var(lnx)$ are the two parameters that determine the shape of the
Lognormal distribution and correspond to the mean and variance of $lnx$, respectively. The
mean value of the Lognormal distribution is given by

$$\langle x \rangle = \int_0^\infty x\, P_L(x)dx = e^{\mu + \frac{\sigma^2}{2}}, \qquad (6)$$

and the variance given by

$$Var(x) = \int_0^\infty (x - \langle x \rangle)^2\, P_L(x)dx = e^{2\mu + \sigma^2}\left(e^{\sigma^2} - 1\right). \qquad (7)$$

It follows from Eq. *(6)* and *(7)* that the inverse relative variance can be derived from the
following equation

$$e^{\sigma^2} = 1 + \frac{Var(x)}{\langle x \rangle^2} = 1 + \frac{1}{v}. \qquad (8)$$

An example of the Gamma and Lognormal distributions for LWC is shown in Figure 1a. In this
example, both distributions have the same mean $\langle LWC \rangle = 0.5 g/kg$ and also the same inverse
relative variance $v = 3$. Although the general shapes of the two PDFs are similar, they differ
significantly at the two ends: the Gamma PDF is larger than Lognormal PDF over the small values
of LWC, and the opposite is true over the large values of LWC. The Gamma and Lognormal
distributions can also be used to describe the spatial variation of CDNC (Gultepe and Isaac, 2004).
An example is given in Figure 1c, in which the LWC=$0.5 g/kg$, the mean CDNC $\langle N_c \rangle = 50\ cm^{-3}$,
and the inverse relative variance of CDNC $v = 5.0$.
Both Gamma and Lognormal distributions are mathematically convenient. For example,
if any physical process $M(x)$ is a power function of $x$,

$$M(x) = Kx^\beta, \qquad (9)$$

then if $x$ follows the Gamma distribution, the expected value $\langle M(x) \rangle$ is given by

$$\langle M(x) \rangle_G = K \int_0^\infty x^\beta\, P_G(x)dx = \frac{\Gamma(v+\beta)}{\Gamma(v)v^\beta}K\langle x \rangle^\beta,\ \beta > -v. \qquad (10)$$

Similarly if $x$ follows the Lognormal distribution, the expected value of $\langle M(x) \rangle$ is





$$\langle M(x)\rangle_L = K \int_0^\infty x^\beta P_L(x)dx = \left(e^{\sigma^2}\right)^{\frac{\beta^2-\beta}{2}} K\langle x\rangle^\beta. \tag{11}$$

Thus, the expected value of $\langle M(x)\rangle$ can be computed from the analytical solutions above,
instead of a numerical integration over the PDF. However, it is important to note that Eq. *(10)* is
only valid when $\beta > -v$. The Gamma function $\Gamma(v+\beta)$ can run into singular values when $v+$
$\beta<0$. In contrast, Eq. *(11)* is valid for any real value $\beta$. This is one advantage of the Lognormal
distribution over the Gamma distribution.
2.2. Impacts of subgrid cloud variations on warm rain simulations in climate models
As pointed out in Pincus and Klein (2000), the subgrid cloud property variations have
important implications for modeling the nonlinear cloud processes in climate models, such as the
precipitation and radiative transfer processes. Of particular interest to this study is the auto-
conversion process that initializes the warm rain in marine boundary layer clouds. Following
Khairoutdinov and Kogan (2000) ("KK2000" hereafter), the auto-conversion rate is often modeled
in GCMs as a power function of LWC and cloud droplet number concentration (CDNC) as follows

$$\frac{\partial q_r}{\partial t} = C(q_c)^{\beta_q}(N_c)^{\beta_N}, \tag{12}$$

where $\frac{\partial q_r}{\partial t}$ is the rain water tendency due to the auto-conversion process, $q_c$ is the cloud water
mixing ratio in the unit of kg/kg, $N_c$ is the CDNC in the unit of cm$^{-3}$. The three parameters $C =$
$1350$, $\beta_q = 2.47$ and $\beta_N = -1.79$ are derived through a least-square fitting of the rain rate
results from a large-eddy simulation. The KK2000 scheme has been adopted in the popular two-
moment cloud microphysics scheme for GCMs developed by Morrison and Gettelman *(2008)*
(referred to as MG scheme). Ideally, if the subgrid variations of $q_c$ and $N_c$ are known, then the
grid-mean in-cloud auto-conversion rate can be derived from the following integral

$$\langle \frac{\partial q_r}{\partial t}\rangle = \int_0^\infty \int_0^\infty C(q_c)^{\beta_q}(N_c)^{\beta_N} P(q_c, N_c)dq_c dN_c, \tag{13}$$

where $P(q_c, N_c)$ is the joint PDF of $q_c$ and $N_c$. Unfortunately, most conventional GCMs lack the
capability of predicting the subgrid variations of cloud properties, with only a couple of
exceptions (Thayer-Calder et al., 2015). What is known from the GCM is usually the in-cloud
grid-mean values $\langle q_c\rangle$ and $\langle N_c\rangle$. As a result, instead of using Eq. *(13)*, the auto-conversion rate
in GCMs is usually computed from the following equation



$$\langle \frac{\partial q_r}{\partial t} \rangle = E \cdot C (\langle q_c \rangle)^{\beta_q} (\langle N_c \rangle)^{\beta_N}, \tag{14}$$

where $E$ is referred to as the "enhancement factor" in Morrison and Gettelman *(2008)*, or the
"subgrid scale homogeneity bias" in Pincus and Klein (2000). By definition its value is the ratio

$$E = \frac{\int_0^\infty \int_0^\infty (q_c)^{\beta_q}(N_c)^{\beta_N} P(q_c, N_c) dq_c dN_c}{(\langle q_c \rangle)^{\beta_q}(\langle N_c \rangle)^{\beta_N}}. \tag{15}$$

The root of this enhancement factor is that the auto-conversion process is a non-linear function
of $q_c$ and $N_c$. As a result, the rain rate computed based on the grid-mean values $\langle q_c \rangle$ and $\langle N_c \rangle$
would be biased in comparison with the result from the integral in Eq. *(13)* (Pincus and Klein,
2000).  Obviously, the value of the enhancement factor depends on the subgrid variations of $q_c$
and $N_c$. If clouds are homogenous on the subgrid scale, then $E \sim 1$. The more inhomogeneous
the clouds are, the larger the $E$ is. In the special case where $q_c$ and $N_c$ are independent, then the
joint PDF $P(q_c, N_c)$ becomes $P(q_c, N_c) = P(q_c)P(N_c)$ , where $P(q_c)$ and $P(N_c)$ are the PDF of
the subgrid $q_c$ and $N_c$. Consequently, Eq. *(13)* reduces to

$$\langle \frac{\partial q_r}{\partial t} \rangle = C \int_0^\infty (q_c)^{\beta_q} P(q_c) dq_c \int_0^\infty (N_c)^{\beta_N} P(N_c) dN_c. \tag{16}$$

And Eq.*(15)* reduces to

$$E = E_q \cdot E_N, \tag{17}$$

where $E_q$ is the enhancement factor due to the subgrid variation of cloud water which has the
form,

$$E_q = \frac{\int_0^\infty \int_0^\infty (q_c)^{\beta_q} P(q_c) dq_c}{(\langle q_c \rangle)^{\beta_q}}, \tag{18}$$

and the $E_q$ is the enhancement factor due to the subgrid variation of cloud water which has the
form,

$$E_N = \frac{\int_0^\infty \int_0^\infty (N_c)^{\beta_N} P(N_c) dN_c}{(\langle N_c \rangle)^{\beta_N}}. \tag{19}$$






Because most current GCMs do not have the capability to simulate the subgrid cloud
property variations, models usually use pre-defined subgrid cloud variations in the computation
of grid-mean auto-conversion rate instead of using prognostic values. For example, in the MG
scheme for the CAM5.3, the subgrid LWC is assumed to follow the Gamma distribution in Eq. *(1)*.
Furthermore, it is assumed that the subgrid variation of CDNC is small and therefore the
enhancement factor due to CDNC variation is negligible (i.e., close to unity). Substituting the
Gamma distribution in Eq. *(1)* into the definition equation of enhancement factor in Eq.(*18*), and
with help from Eq. *(10)*, one can derive that

$$E(P_G, \beta) = \frac{1}{\langle x \rangle^\beta} \int_0^\infty x^\beta \, P_G(x) dx = \frac{\Gamma(v+\beta)}{\Gamma(v)v^\beta}, \qquad (20)$$

where $x \sim q_c, \beta = \beta_q = 2.47$ for the enhancement factor for the KK2000 scheme due to the
subgrid variation of cloud water. In addition to the Gamma distribution, some studies also use
the Lognormal distribution to account for the subgrid cloud water variation (Lebsock et al., 2013).
In such case, substituting the Lognormal distribution in Eq. *(5)* into Eq.(*18*), and with help from
Eq.*(11)*, one can find that the enhancement factor for the Lognormal distribution is given by

$$E(P_L, \beta) = \frac{1}{\langle x \rangle^\beta} \int_0^\infty x^\beta \, P_L(x) dx = \left(e^{\sigma^2}\right)^{\frac{\beta^2-\beta}{2}} = \left(1+\frac{1}{v}\right)^{\frac{\beta^2-\beta}{2}}. \qquad (21)$$

Figure 1b shows the rain rate based on the KK2000 parameterization scheme for the
Gamma and Lognormal LWC PDF in Figure 1a. Interestingly, although the cumulative rain rates
based on the two types of PDFs are almost identical, the contribution to the total rain rate from
the different LWC bins are quite different. As show in Figure 1a, the $P_L(q_c)$ has a longer tail than
the $P_G(q_c)$, i.e., the occurrence probability of large LWC (e.g., $q_c > 2.0 g/kg$ ) is much higher in
the Lognormal than in Gamma PDF. This difference is further amplified in the rain rate
computation in Figure 1b because the rain rate is proportional to $q_c^{2.47}$.
The enhancement factors based on the Gamma (i.e., $E(P_G, \beta)$ in Eq. *(20)*) and Lognormal
(i.e., $E(P_L, \beta)$ in Eq. *(21)*) PDF for $\beta_q = 2.47$ are plotted as a function of the inverse relative
variance $v$ in Figure 2. When subgrid clouds are more homogenous i.e., $v > 1$, the enhancement
factor based on the two PDFs are similar. However, for more inhomogeneous grids with i.e., $v <$
1, the $E(P_L, \beta)$ is significantly larger than that $E(P_G, \beta)$, which is probably because of the longer
tail of $P_L(q_c)$ as shown in Figure 1 a and b.



It is important to note that not only the subgrid variation of $q_c$ can lead to a nonlinear

effect on the simulation of autoconversion rate, the subgrid variation of $N_c$ can have the same
effect. Physically, provided the same LWC, a cloud with smaller $N_c$ would have larger droplet size
and therefore larger precipitation efficiency than the cloud with larger $N_c$. Because the
autoconversion rate depends nonlinearly on $N_c$, the grid-mean autoconversion rate computed
based on a skewed PDF of $N_c$ (i.e., $\int_0^\infty (N_c)^{\beta_N} P(N_c) dN_c$) would be different from that computed
based on the mean of $N_c$ (i.e., $(\langle N_c \rangle)^{\beta_N}$). The autoconversion enhancement factor based on the
Lognormal PDF $E(P_L, \beta)$ for $\beta_N = -1.79$ is given in Figure 2. Interestingly, at the same inverse
relative variance $v$, the enhancement factor based on the same Lognormal PDF $E(P_L, \beta)$ for $\beta_N =$
$-1.79$ is actually larger than that for $\beta_q = 2.47$ because of the formula of the exponent in Eq.
*(21)* (i.e., $\frac{\beta^2 - \beta}{2}$). This potentially important effect of the subgrid inhomogeneity of $N_c$ on the
simulation of autoconversion rate has been overlooked or ignored in most previous studies. It is
perhaps partly because modeling $N_c$ in GCM, especially its subgrid variation, is notoriously
difficult, and also partly because there is a lack of observation-based study of the subgrid
variation of $N_c$. One important objective of this study is to fill the second gap. We will use MODIS
observations to investigate the role of subgrid $N_c$ variation on autoconversion simulation.

Finally, it has to be noted that when both $q_c$ and $N_c$ have significant subgrid variations,

their covariation also becomes important. As explained in Griffin and Larson (2013), if the $q_c$ and
$N_c$ are negatively correlated, clouds with larger $q_c$ would tend to have smaller $N_c$. The
autoconversion rate in such a case would be larger than that in the case where $q_c$ and $N_c$ are
positively correlated (i.e., larger $q_c$ would tend to have larger $N_c$). As explained in Eq. *(17)*, only
when they are uncorrelated can the total enhancement factor be decomposed into the product
of two independent factors $E = E_q \cdot E_N$. Otherwise additional terms are necessary to take into
account the effect of $q_c$ and $N_c$ correlation. Although potentially important, the correlation of $q_c$
and $N_c$ from satellite remote sensing data is difficult to derive from the satellite remote sensing
observations due to the retrieval uncertainties. We will return to this point later in Section 5.3.

3.  Data and Methodology

Of particular interest to this study are the grid-mean value and subgrid variation of several



key properties of liquid-phase clouds, namely, COT, CER, LWP and CDNC, in the tropical regions.
For this purpose, we use the latest collection 6 (C6) *daily mean* level-3 cloud retrieval product
from the Aqua-MODIS instrument (product name "MYD08_D3"). The MODIS level-3 (i.e., grid-
level) product contains statistics computed from a set of level-2 (i.e., pixel-level) MODIS granules.
As summarized in (Platnick et al., 2003; 2017), the operational level-2 MODIS cloud product
provides cloud masking (Ackerman et al., 1998), cloud top height (Menzel et al., 1983), cloud top
thermodynamic phase determination (Menzel et al., 2006), and COT, CER and LWP retrievals
based on the bi-spectral solar reflectance method (Nakajima and King, 1990). All MODIS level-2
atmosphere products, including the cloud, aerosol and water vapor products, are aggregated to
1°×1° spatial resolution on a daily, eight-day, and monthly basis. Aggregations include a variety
of scalar statistical information, including mean, standard deviation, max/min occurrences, as
well as histograms including both marginal and joint histograms. For COT, CER and LWP, the
MODIS level-3 product provides both their "in-cloud" grid-mean values ($\langle x \rangle$) and subgrid
standard deviations ($\sigma_x$). The inverse relative variance $v$ can then be derived from Eq. *(4)*, i.e.,
$v = \langle x \rangle^2 / \sigma_x^2$. Note that the operational MODIS product provides two CER retrievals, one based
on the observation from the band 7 centered around 2.1 μm and the other from band 20 at 3.7
μm. As discussed in several previous studies (Cho et al., 2015; Zhang and Platnick, 2011; Zhang
et al., 2012; 2016), the 3.7 μm band CER retrieval is more resilient to the 3-D effects and retrieval
failure than the 2.1 μm band retrievals. For these reasons, it is used as the observational
reference in this study.

Given the COT and CER retrieval, the operational MODIS product estimates the LWP of cloud

using

$$LWP = \frac{2}{3} \rho_w COT \cdot CER, \tag{22}$$

where $\rho_w$ is the density of water. Several studies have argued that a smaller coefficient of 5/9,
instead of 2/3, should be used in estimation of LWP (Seethala and Horváth, 2010; Wood and
Hartmann, 2006). The choice of the coefficient has no impact on our study, because we are
interested in the relative inverse variance $v = \langle x \rangle^2 / \sigma_x^2$. We note here that it is the LWC, instead
of the LWP, that is used in the KK2000 scheme. So, the spatial variability of LWC is what is most





relevant. However, the remote sensing of cloud water vertical profile from satellite sensor for
liquid-phase clouds is extremely challenging even with active sensors. It is why most previous
studies using the satellite observations analyzed the spatial variation of LWP, rather than LWC.
In fact, even Lebsock et al. (2013), who used the level-2 CloudSat observations, had to use the
vertical averaged LWC in their analysis. Ground-based observations are much better than
satellite observation in this regard because they are closer to the target (i.e. clouds). Recently,
Xie and Zhang (2015) analyzed the cloud water profiles retrieved using ground-based radars from
the three ARM sites and found no obvious in-cloud vertical dependence of the spatial variability
of LWC. Following these previous studies, we assume that the horizontal subgrid variation of LWC
is *not* strongly dependent on height and its value can be inferred from the spatial variability of
the vertical integrated quantity LWP. The uncertainty caused by this assumption will be assessed
in future studies.

The current MODIS level-3 cloud product does *not* provide CDNC retrievals. Following

previous studies (Bennartz, 2007; Bennartz and Rausch, 2017; Grosvenor and Wood, 2014;
McCoy et al., 2017a), we estimate the CDNC ($N_c$) of liquid-phase clouds from the MODIS retrieved
COT ($\tau$) and CER ($r_e$) based on the classic adiabatic cloud model

$$N_c(\tau, r_e) = \frac{\sqrt{5}}{2\pi k} \frac{\sqrt{f_{ad}\Gamma_w}}{\sqrt{\rho_w Q_e}} \tau^{\frac{1}{2}} r_e^{-\frac{5}{2}}, \tag{23}$$

where $\rho_w$ is the density of water; $Q_e \approx 2$ is the extinction efficiency of cloud droplets; $k$ is the
ratio of $r_e$ to mean volume-equivalent radius; $f_{ad}$ is the adiabaticity of the cloud; $\Gamma_w$ is the LWC
lapse rate. Following previous studies, we assume $k = 0.8$ and $f_{ad} = 1.0$ to be constant and
compute  $\Gamma_w$ from the grid mean liquid cloud top temperature and pressure.  The theoretical
basis and main uncertainty sources of the CDNC estimation based on the adiabatic cloud model
from MODIS-like passive cloud retrievals are nicely reviewed by Grosvenor et al. (2018).

Ideally, the values of $LWP$ and CDNC should be estimated on pixel-by-pixel basis from the

level-2 MODIS product. However, pixel-by-pixel estimation is highly time consuming, which
makes it difficult to achieve a global perspective. Using an alternative method, many previous
studies estimate the grid-level CDNC statistics from the joint histogram of COT vs. CER provided



in the level-3 MODIS cloud products (Bennartz, 2007; McCoy et al., 2017a; 2017b). For a given
1°×1° grid-box, the liquid-phase COT-CER joint histogram provides the counts of successful cloud
property retrievals with respect to 108 joint COT-CER bins that are bounded by 13 COT bin
boundaries, ranging from 0 to 150, and 10 CER bin boundaries, ranging from 4 μm to 30 μm. With
the joint histogram, which is essentially the joint PDF of COT and CER $P(\tau, r_e)$, we can estimate
the grid mean and variance of CDNC from the following equations

$$\langle x \rangle = \int \int x(\tau, r_e) P(\tau, r_e) d\tau dr_e, \tag{24}$$

$$Var(x) = \int \int (x(\tau, r_e) - \langle N_c \rangle)^2 P(\tau, r_e) d\tau dr_e, \tag{25}$$

where $x$ can be either LWP or CDNC. Figure 3a shows the LWP in Eq. *(22)* as a function of the 13
COT bins and 10 CER bins from the MODIS level-3 product. As expected, the largest LWP values
are found when both COT and CER are large. Figure 3b shows the CDNC in Eq. *(23)* as a function
of the COT and CER bins. As expected, the largest CDNC values are found when both COT is large
and CER is small. Figure 3c shows an example of the COT-CER joint histogram from the Aqua-
MODIS daily level-3 product "MYD08_D3" on January 09[th], 2007 at the grid box 1°S and 1°W. In
this particular grid box, a combination of 2~4 COT and 10 μm ~12 μm CER is the most frequently
observed cloud value. Using the joint histogram in Figure 3c, we can derive the mean and variance
of both LWP and COT using the Eqs. *(24)* and *(25)*.

The efficiency of using the level-3 product is accompanied by two important limitations.

First, the current level-3 MODIS cloud product has a fixed 1°x1° spatial resolution. Although this
resolution is highly relevant to the current generation of GCMs, i.e., CMIP5 (Taylor et al., 2012),
future GCMs may have significantly finer resolution. Second, it is difficult to sub-sample the pixels
with the best retrieval quality. As reviewed in Grosvenor et al. (2018), the main source of
uncertainty in the CDNC retrieval is the MODIS retrieval uncertainties, particularly in CER because
of $N_c \sim r_e^{-\frac{5}{2}}$ dependence. In the pixel-by-pixel method, the pixel-level retrieval uncertainties, as
well as some other metrics such as the sub-pixel inhomogeneity index, provided in the level-2
product can be used to select the pixels with the best retrieval quality. Here, because we use the
static COT-CER joint histogram provided in the operational level-3 product, we do not have the





flexibility to sub-sample the data using retrieval quality. Alternatively, we can sub-sample the
data using the COT. It is well known that the bi-spectral retrieval method has a large uncertainty
for thin clouds. Indeed, the clouds with COT thinner than about 4 have often been screened out
in previous studies (Quaas et al., 2008). Such screening can be easily done with the joint PDF of
COT and CER, but it would obviously lead to sampling bias in LWP. The impact on CDNC is
dependent on whether the CDNC is correlated with the COT, i.e., whether thin clouds have the
similar CDNC as the thick clouds. We will revisit this point later. It should be noted that because
thin clouds in MODIS retrieval tend to have large uncertainty, any type of data quality-based data
screening would inevitably lead to the sampling bias.
4.  Grid-mean and subgrid variations of liquid-phase cloud properties

The annual mean total cloud fraction ($f_{tot}$), liquid-phase cloud fraction ($f_{liq}$), in-cloud COT,

CER from the 3.7 μm band, LWP and estimated CDNC over the tropical oceans based on 10 years
Aqua-MODIS retrievals are shown in Figure 4. The highest $f_{liq}$ in the tropics is usually found in the
stratocumulus (Sc) decks over the Eastern boundary of the ocean, e.g., SE Pacific off coast of Peru,
NE Pacific off the coast of California and SE Atlantic off the coast of Namibia. These regions are
associated with relatively low sea surface temperature (SST) due to cold upwelling ocean surface
current and mid-tropospheric subsidence of warm air from large-scale circulations, which
together lead to a strong low-tropospheric stability and high liquid-cloud fraction. With an annual
mean TOA cloud radiative effect usually around −40 ~ −60 W/m², the Sc decks are important
modulators of the local and global radiative energy budget. The liquid-cloud fraction reduces
significantly toward the open ocean trade wind regions, where the dominate cloud types are
broken cumulus (Cu). Close to the continents, the Sc decks are susceptible to the influence of
continental air mass with higher loading of aerosols in comparison with pristine ocean
environment, which is probably the reason the SC decks have smaller CER and higher CDNC than
the open-ocean trade cumulus (Figure 4 d and f). The in-cloud COT (Figure 4 c) and LWP (Figure
4 e) generally increase from the Sc decks to the open-ocean Cu regime, although less dramatically
than the transition of cloud fraction. The Sc decks and the Sc-to-Cu transition are the most
prominent features of liquid-phase clouds in the tropics. However, as mentioned in the
introduction, simulating these features in the GCMs proves to be an extremely challenging task,





and most GCMs suffer from some common problems, such as the "too few too bright" problem
and the abrupt Sc-to-Cu transition problem (Kubar et al., 2014; Nam et al., 2012; Song et al.,

2018).

Switching the focus now from grid-mean values to subgrid variability, we show the grid-

level inverse relative variances $v = \langle x \rangle^2 / Var(x))$ for several key cloud properties. Recall that $v$
is defined such that the larger the $v$, the larger the mean value in comparison with the variance,
and the more homogeneous the cloud property within the grid. Because the value of $v$ can be ill-
behaved when $Var(x)$ approaches zero, instead of the mean value, we plot the median value of
$\tilde{v}$ based on 10 years of MODIS observations in Figure 5. There are several interesting and
important features in Figure 5. First of all, the $\tilde{v}$ of all four sets of cloud properties (i.e., COT, CER,
LWP and CDNC) all exhibits a clear and similar Sc-to-Cu transition, with larger values in the Sc
region and smaller value in the broken Cu regions. This indicates that cloud properties, including
both optical and microphysical properties, are more homogenous, in terms of spatial distribution
within the grid, in the Sc region than in the Cu region. Secondly, the value of $\tilde{v}$ of CER (i.e., 10~100
in Figure 5b) is larger than that of the other properties (i.e., 1~10) by almost an order of
magnitude, indicating that the subgrid variability of CER is very small. On the hand, however, it is
important to note that the $\tilde{v}$ of CDNC (Figure 5d) is comparable with that of COT (Figure 5a) and
LWP (Figure 5c). The reason is probably in part because the highly nonlinear relationship between
CDNC and CER (i.e., $N_c \sim r_e^{-\frac{5}{2}}$ ) leads to a stronger variability of CDNC than CER, and also in part
because the variability of CDNC is also contributed by the subgrid variation of COT.  In some
regions, the Gulf of Guinea, East and South China Sea, and Bay of Bengal for example, the $\tilde{v}$ of
CDNC is close to unity, indicating the subgrid standard deviation of CDNC is comparable to the
grid-mean values in these regions. As discussed in the next section, the significant subgrid
variability of CDNC in these regions should be taken into account when modeling the nonlinear
processes, such as the auto-conversion, in GCM to avoid systematic biases due to the nonlinearity
effect.

The values of $\tilde{v}$ in Figure 5 from this study are in reasonable agreement with previous

studies. Barker (1996) selected a few dozens of cloud scenes, each about 100 ~ 200 km in size,
from the Landsat observation and analyzed their spatial variability of COT. It is found that the



typical value of $v$ for "overcast stratocumulus", "broken stratocumulus" and "scattered cumulus"
is 7.9, 1.2, and 0.7, respectively (see their Table 3), which is consistent with the Sc-to-Cu transition
pattern seen in Figure 5. Oreopoulos and Cahalan (2005) derived the subgrid inhomogeneity of
COT on a global scale from the level-3 Terra-MODIS retrievals. Although using a different metric
(i.e., their inhomogeneity parameter is defined as $\chi = \exp(\ln\langle\tau\rangle)/\langle\tau\rangle$), they also found
systematic increase of inhomogeneity (decreasing value of $\chi$) from the Sc region to cu region.
Also using the MODIS cloud property retrievals, Wood and Hartmann(2006) investigated the
meso-scale spatial variability of LWP in the NE Pacific and SE Pacific region. The $v$ of LWP is found
to increase systematically with meso-scale cloud fraction and the relationship between the two
can be reasonably explained by a simple PDF cloud thickness model in Considine et al. (1997).
See also Kawai and Teixeira (2010).
5.   Implications for warm-rain simulations in GCM
5.1. Influence of subgrid variation of LWP
As explained in the Theoretical Background, in GCMs the influences of subgrid cloud water
variability on the simulation of highly nonlinear autoconversion process are accounted for using
the enhancement factors defined in Eq. *(15)*. For example, in CAM5.3, the MG cloud microphysics
parameterization scheme assumes that the subgrid cloud water follows the Gamma distribution
with the value of $v = 1$, which leads to a constant enchantment factor of 3.2 for the KK2000
autoconversion scheme (Morrison and Gettelman, 2008). Because its direct connection with the
precipitation rate, the enhancement factor can have significant impacts on precipitation, cloud,
and radiation fields of the host model. For the same reason, it is also often used as a "tuning"
parameter to optimize the model and reduce the differences between model simulations and
observations (Guo et al., 2014). Thus, an observational constraint on the enhancement factor is
of great interest to the modeling community and has been the target of several recent studies.
In the part 1 of a two-part study, Larson and Griffin (2013) present a theoretical framework based
on the joint PDF of cloud and meteorological properties for diagnosing the enhancement factors
for various nonlinear processes in warm clouds, e.g., autoconversion, accretion, and evaporation.
In part 2, Griffin and Larson (2013) analyzed the in situ measurements from the research flight
two (RF02) of the second Dynamics and Chemistry of Marine Stratocumulus (DYCOMS-II) field



experiment. It is found that taking into account the nonlinear effect caused by subgrid cloud
variability increases the autoconversion and accretion rates, leading to significantly more surface
precipitation and better agreement to the observations.
As discussed in Section 2.2, given the subgrid cloud property variations, we can derive the
enhancement factor using two approaches. In the first, we can derive the enhancement factor
based on its definition in Eq. *(18)* and *(19)* directly from the observed PDF of LWP or CDNC,
respectively. The advantage of this approach is that we do not have to make any assumption
about the shape of the subgrid cloud property variation (i.e., Gamma or Lognormal), although
this approach is more time consuming because it has to solve the integration. In the second
approach, we first derive the relative inverse relative variance $v$ and then derive the
enhancement factor by assuming the subgrid PDF to be either Gamma (i.e., Eq. *(20)*) or
Lognormal (i.e., using Eq. *(21)*). This approach is more although it may be subject to significant
error if the true PDF deviates from the assumed PDF shape.
Figure 6a shows the median enhancement factor $E_q$ in the tropical region derived based
on Eq. *(18)* (i.e., the first approach) from 10 years of MODIS observation. Figure 6 b and c show
the median enhancement factor $E_q$ derived by assuming the subgrid cloud water follows the
Lognormal and Gamma distribution, respectively. There are a couple of interesting and important
points to note. First of all, similar to the grid-mean quantities in Figure 4, the enhancement factor
$E_q$ also shows a clear Sc-to-Cu transition. Over the Sc decks, because clouds are more
homogeneous ($\tilde{v} > 5$), the enhancement factor $E_q$ is only around 1 ~ 2.5, while over the Cu
regions, the more inhomogeneous clouds with $\tilde{v} < 1$ leads to a larger enhancement factor $E_q$
around 3~5. As aforementioned, in the current CAM5.3, $E_q$ is assumed to be a constant of 3.2.
While this value is within the observational range, it obviously cannot capture the Sc-to-Cu
transition. In fact, the constant value 3.2 overestimates the $E_q$ over the Sc region and
underestimates the $E_q$ over the Cu region, which could lead to unrealistic drizzle product in both
regions and to consequential impacts on cloud water budget, radiation and even aerosol indirect
effects on the model. The second point to note is that the $E_q$ based on the Lognormal PDF
assumption in Figure 6 b agrees well with the results in Figure 6 b derived directly from the
observation. In contrast, the $E_q$ based on the Gamma PDF assumption in Figure 6 c tend to be





smaller, especially in the Cu regions. This result seems to suggest that the Lognormal distribution
provides a better fit to the observed subgrid cloud water variation than the Gamma distribution,
which has rarely been noted and reported in the previous studies.

A flexible, cloud-regime dependent $E_q$ could help improve the simulation of Sc-to-Cu

transition in the GCM. If a GCM employs an advanced cloud parameterization scheme, such as
CLUBB, that is able to provide regime-dependent information on subgrid cloud variation, i.e., $v$,
then the enhancement factor $E_q$ could be diagnosed from $v$ . However, most traditional cloud
parameterization schemes do not provide information on subgrid cloud variation. In such case, if
one does not wish to use a constant $E_q$, but a varying regime-dependent scheme, then either $v$
or $E_q$ need to be parameterized as a function of some grid-mean cloud properties resolved by
the GCM. In facts, several attempts have been made along this line. Based on the combination
air-borne in situ measurement and satellite remote sensing product, Boutle et al. (2014)
parameterized the "fractional standard deviation" (which is equivalent to $1/\sqrt{v}$ in our definition)
of liquid-phase cloud as a function of grid-mean cloud fraction. This scheme was later updated
and tested in a host GCM in Hill et al. (2015), and was found to reduce the shortwave cloud
radiative forcing biases in the model. In a recent study, Xie and Zhang (2015) derived the subgrid
cloud variations from the ground-based observations from three Department of Energy (DOE)
Atmospheric Radiation Measurement (ARM) sites, and then parameterize the inverse relative
variance $v$ as a function of the atmospheric stability.

Figure 7a shows the variation of inverse relative variance $v$ as a function of the grid-mean

liquid-phase cloud fraction $f_{liq}$. In general, the value of $v$ increases with the increasing $f_{liq}$, which
is expected from the Sc-to-Cu increase of $f_{liq}$ in Figure 4b and the Sc-to-Cu decrease of $v$ in Figure
5c. The $v(f_{liq})$ pattern in Figure 7a is also consistent with the results reported in Wood and
Hartmann (2006) and Lebsock et al. (2013). In the hope of obtaining a simple parameterization
scheme for $v(f_{liq})$ that can be used in GCMs, we fit the median value of $v$ as a simple 3rd order
polynomial of $f_{liq}$ as follows:

$$v(f_{liq}) = 2.38 - 4.95 f_{liq} + 8.74 f_{liq}^2 - 0.49 f_{liq}^3, \; f_{liq} \in (0,1]. \tag{26}$$



To test the performance of this simple parameterization, we first substitute the $f_{liq}$ from MODIS
daily mean level-3 product into the above equation and then use the resultant $v$ to compute the
enhancement factor $E_q$. Unfortunately, the median value of the enhancement factor $E_q$
computed based on the parameterized $v(f_{liq})$ as shown in Figure 8a substantially underestimate
the observation-based results in Figure 6, especially over the Cu regions. The deviation is
probably because the relationship between $E_q$ and $v$ is highly nonlinear (e.g., Eq. *(20)* and *(21)*)
and therefore the above parameterization scheme that only fits the median value of $v$ is not able
to capture the variability of $E_q$. Based on this consideration, we tried an alternative approach.
Instead of parameterization of $v$, we directly parameterize the enhancement factor $E_q$ as a
function of $f_{liq}$. Figure 7b shows the variation of $E_q$ as a function of $f_{liq}$. As expected, $E_q$ generally
decreases with increasing $f_{liq}$. The median value of $E_q$ is fitted with the following 3$^{\text{rd}}$ order
polynomial of $f_{liq}$

$$E_q(f_{liq}) = 2.72 + 7.33 f_{liq} - 19.17 f_{liq}^2 + 10.69 f_{liq}^3, \ f_{liq} \in (0,1]. \tag{27}$$

As shown in Figure 8b, the median value of $E_q$ based on the above equation clearly agrees with
the observation-based values in Figure 6 better than that based on the parameterization of
$v(f_{liq})$. The elimination of the middle step indeed improves the parameterization results. While
this is encouraging, it should be kept in mind that the Eq. *(27)* has very limited application, i.e., it
is only useful for the autoconversion rate computation for a particular value of the
autoconversion exponent beta, i.e., $\beta_q = 2.47$. A good parameterization of $v$ could be useful for
not only autoconversion, but also for accretion and radiation computations. Another caution is
that, if applied to a GCM, the performance of the $E_q(f_{liq})$ parameterization in Eq. *(27)* will be
dependent on the simulated accuracy of $f_{liq}$ in the model. In future study, we will implement this
parameterization scheme in a couple of GCMs and study the impacts on the cloud, precipitation
and radiation simulations. We will also explore better ways to parameterize the inverse relative
variance $v$.



5.2. Influence of subgrid variance of CDNC

In the previous section, we have mainly focused on the enhancement factor $E_q$ on

autoconversion simulation due to the subgrid variation of cloud water. In this section we switch
the focus on the enhancement factor $E_N$ due to the subgrid variation of CDNC.

The median value of $E_N$ derived based on Eq. *(19)* from 10 years of MODIS observation is

shown in Figure 9a. There are several intriguing points to note. First of all, the value of $E_N$ is
actually larger than $E_q$ in Figure 9 such that we even have to use a different color scale for this
plot. Secondly, $E_N$ the regions with escalated $E_N$ seem to coincide with the downwind regions of
biomass burning aerosols (e.g., Gulf of Guinea, East Coast of South Africa), air pollution (i.e.,
Eastern China Sea), and, most interestingly, active volcanos (e.g., Kilauea Hawaii and Ambae
Vanuatu). We have also checked the seasonal variation of the $E_N$ (shown in supplementary
materials) and the results also support this observation. Another interesting feature to note is
that, although the dust outflow regions such as Tropical East Atlantic and Arabian Sea, have heavy
aerosol loading, the value of $E_N$ there is only moderate. Figure 9b shows the value of $E_N$
computed based on Eq. *(21)* from the inverse relative variance of $v$, assuming that the subgrid
CDNC follows a Lognormal PDF. Although the overall pattern is consistent with Figure 9a, the
assumption of Lognormal PDF seems to underestimate $E_N$. A closer examination indicates that
the Lognormal PDF tend to underestimate the population of clouds with small CDNC, and
therefore underestimate the variance of CDNC as well as $E_N$. We did not compute the $E_N$ based
on the Gamma distribution because of the singular value problem aforementioned in Section2.1.

We could not find any previous observation-based study on the global pattern of the

subgrid variation of CDNC and the corresponding $E_N$. So, it is difficult for us to corroborate our
results. On one hand, the pattern of $E_N$ in Figure 9a seems to suggest that there are some
underlying physical mechanisms controlling the subgrid variation of CDNC, in which aerosols
seem to play an important role. On the other hand, the magnitude of $E_N$ is surprisingly large. As
explained in section 3, the CDNC is estimated based on Eq. *(23)* from the MODIS retrieval of COT
and CER. Could retrieval uncertainty contribute to the large subgrid variation of CDNC and
therefore $E_N$? In order to better understand the large value of $E_N$, we selected a case during the
biomass burning season in the Gulf of Guinea, which is shown in Figure 10. During the boreal



winter, the grassland and savanna fires in the southern West Africa generate a thick layer of
smoke aerosols that are clearly visible in the satellite image (Andreae and Merlet, 2001). On this
day, the Gulf of Guinea is quite cloudy, filled with broken cumulus clouds in the northern coastal
region and stratiform clouds in the south. We arbitrarily selected a smaller region, marked with
the red box, for detailed analysis. Although the cloud fraction in this region is about 60%, the
clouds are broken and optically thin with COT mostly smaller than 10. Interestingly, the CER varies
substantially from as low as 4 μm up to 30 μm in this relatively small region. Because of the highly
nonlinear dependence of CDNC on CER (i.e., $N_c \sim r_e^{-5/2}$), the large variance of CER leads to an
even larger variance of CDNC. The $E_N$ derived based on Eq. *(19)* is 9.9. In contrast, the $E_q$ is only
about 1.5.

The results from the above case study raises some concerns. It seems that the large

variations of CER and therefore CDNC are usually associated with thin clouds. While there could
be a physical explanation (e.g., CCN activation), it seems more likely to be caused, or at least
contributed, by retrieval uncertainty. It is well known that the bispectral method has large
uncertainties for thin clouds, especially when they are broken. Several previous studies have
shown that the sub-pixel level surface contamination, subpixel inhomogeneity, and three-
dimensional radiative transfer effects, tend to cause overestimated CER retrieval on top of large
uncertainties (Zhang and Platnick, 2011; Zhang et al., 2012; 2016). Therefore, for such a
challenging case in Figure 10, it is not surprising that the large CDNC variation and $E_N$ are partly
caused by retrieval uncertainty. Based on this consideration, we did a sensitivity test, in which
we screen out the thin clouds with COT < 4 in the computation and analysis of CDNC and $E_N$. The
result from this test is shown in Figure 9c. Indeed, the removal of thin clouds substantially
reduces the value of $E_N$. For example, in the Gulf of Guinea, the median value of $E_N$ reduces by
a factor of 4 from about 10 to only about 2.5. Nevertheless, the global pattern of $E_N$ still remains,
i.e., nonnegligible values of $E_N$ are found in the downwind regions of biomass burning, air
pollution and volcano emission.

As far as we know, the results in Figure 9 and Figure 10 mark the first attempt based on

satellite observations to unveil the global pattern of the subgrid variations of CDNC and
investigate the consequential impacts on warm rain simulations in GCMs. Although obscured by





satellite retrieval uncertainties, the results still provide several valuable insights. First of all, the
enhancement factor $E_N$ due to the subgrid variations of CDNC is nonnegligible, even comparable
the effect of subgrid cloud water variation (i.e., $E_q$). Second, the global pattern of $E_N$ in Figure 9
provides a valuable map for future studies, which in our opinion should focus on the regions with
large $E_N$, e.g., Gulf of Guinea, East Coast of South Africa and Eastern China Sea. Last, but not least,
the example in Figure 10 clearly exposes the limitation of the current satellite remote sensing
method. There are alternative methods for retrieving the CDNC from satellite observations (see
discussion in Grosvenor et al. (2018)). However, these methods more or less face the same
challenges as the MODIS retrieval (i.e., surface contamination, 3D effects). Future studies should
consider using the air-borne in situ measurements of cloud microphysics in the regions with
significant $E_N$, if available.

5.3. The combined effect of subgrid variations of cloud water and CDNC

As discussed in Section 2.2, the combined effect of the subgrid variations of cloud water

and CDNC can be derived from joint PDF $P(q, N_c)$ based on Eq. *(15)*. Because both $q$ and $N_C$ are
a function of the retrieved COT and CER, we can easily derive the combined enhancement factor
$E$ from the COT-CER joint histogram of MODIS product simply changing the integration domain
of Eq. *(15)* from $q$ and $N_C$ to COT and CER. The median value of the combined enhancement
factor $E$ based on Eq. *(15)* is shown in Figure 11a. As one would expect, the combined
enhancement factor is generally larger than both $E_q$ in Figure 6 and the $E_N$ in Figure 9. It is easy
to see that the in some regions (e.g., Gulf of Guinea, East Coast of South Africa and Eastern China
Sea) the combined enhancement factor $E$ resembles the $E_N$ while in other regions (i.e., trade
wind cumulus regions over open ocean) it resembles more of $E_q$. Interestingly, because both $E_q$
and $E_N$ are small over the Sc decks, those regions have the smallest combined enhancement
factor $E$.

As discussed in Section 2.2, only when the subgrid variation of cloud water is uncorrelated

with the subgrid variation of CDNC can the combined enhancement factor $E$ be decomposed into
the simple product of $E_q$ and $E_N$ (i.e., Eq. *(17)*). Otherwise, additional terms that could be quite
complicated are needed to account for the effect of correlation (Lebsock et al., 2013). Here, we
performed a couple of simple tests to understand the potential correlation between $E_q$ and $E_N$.



In the first test, we simply compare the product $E_q \cdot E_N$ with the observation-based $E$ in Figure
11a and we found that the simple product $E_q \cdot E_N$ substantially overestimates $E$, especially over
the region with large $E_N$ (not shown). In the light of the example in Figure 10, in the second test
we screened out the optical thin clouds and computed the $E_q \cdot E_N (COT > 4)$, which is shown in
Figure 11b. It should be clarified that optically thin clouds are kept in the computation of both $E_q$
and $E$, only left out in $E_N$. Apparently, the $E_q \cdot E_N (COT > 4)$ agrees reasonably well with the
combined enhancement factor in Figure 11a. This is encouraging on one hand, but on the other
not easy to explain. A possible explanation is that there is an apparent positive correlation
between cloud water and CDNC in the region with large $E_N$ (i.e., optically thin clouds with less
cloud water tend to have larger CER and smaller CDNC). This correlation mainly exists among
optically thin clouds as a result of retrieval bias and uncertainty and it tends to counteract the
effect of $E_q$ and $E_N$ making the combined enhancement factor $E$ substantially smaller than the
simple product of $E_q \cdot E_N$ (i.e., assuming no correlation).

6.  Summary and Outlook

One of the difficulties in GCM simulation of the warm rain process is how to account for

the impact of subgrid variations of cloud properties, such as cloud water and CDCN, on nonlinear
precipitation processes such as autoconversion. In practice, this impact is often treated by adding
the enhancement factor term to the parameterization scheme. In this study, we derived the
subgrid variations of liquid-phase cloud properties over the tropical ocean using the satellite
remote sensing products from MODIS and investigated the corresponding enhancement factors
for parameterizations of autoconversion rate. In comparison with previous work, our study is
able to shed some new light on this problem in the following regards:

1.  The wide spatial coverage of the Level-3 MODIS product enables us to depict a

detailed quantitative picture of the enhancement factor $E_q$ due to the subgrid

variation of cloud water, which shows a clear cloud regime dependence, i.e., a Sc-

to-Cu increase. The constant $E_q = 3.2$ used in the current CAM5.3 model

overestimates and estimates the observed $E_q$ in the Sc and Cu regions,

respectively.



2. The $E_q$ based on the Lognormal PDF assumption performs slightly better than that
based on the Gamma PDF assumption.
3. A simple parameterization scheme is provided to relate $E_q$ to the grid-mean liquid
cloud fraction, which can be readily used in GCMs.
4. For the first time, the enhancement factor $E_N$ due to the subgrid variation of CDNC
is derived from satellite observation, and the results reveal several regions
downwind of biomass burning aerosols (e.g., Gulf of Guinea, East Coast of South
Africa), air pollution (i.e., Eastern China Sea), and active volcanos (e.g., Kilauea
Hawaii and Ambae Vanuatu), where the $E_N$ is comparable, or even larger than
$E_q$, even after the optically thin clouds are screened out.
In future studies, we will further investigate the implications of these findings from
observations for warm rain simulations in GCMs. For example, the parameterization scheme of
$E_q(f_{liq})$ in Eq. ($27$) can be implemented in the GCMs and compared to the results based on the
constant $E_q$ assumption to understand the potential influence of considering a cloud-regime-
dependent $E_q$ on cloud simulations. Recently, a few novel methods have been developed to
provide certain information on the subgrid cloud property variations to the host GCM. Most
noticeable examples are the super-parameterization method (a.k.a. multi-scale modeling
framework) (Wang et al., 2015) and the higher-order turbulence closure methods (e.g., Cloud
Layer Unified By Binormals, CLUBB) (Golaz et al., 2002a; Guo et al., 2015; Larson et al., 2002).
Those GCMs coupled with these new schemes, theoretically, would no longer need the
enhancement factor. Nevertheless, the subgrid cloud property variations derived in this study
provide the observational basis for the evaluation and improvement of these schemes.
As noted in the previous sections, this study has several important limitations, most of
which are a result of using the level-3 MODIS observations. The fixed 1°x1° spatial resolution of
MODIS level-3 product makes it impossible for us to investigate the scale-dependence of subgrid
cloud variation. Similar to previous studies, we have to make several assumptions when
estimating the CDNC from level-3 MODIS product. Furthermore, the retrieval uncertainties
associated with the optically thin clouds in MODIS product pose a challenging obstacle for the
quantification of subgrid cloud property variations and the corresponding enhancement factors.





These limitations have to be addressed using additional independent observations from, for
example, ground based remote sensing product and/or in situ measurement from air-borne field
campaigns. Nevertheless, the results from this study provide a valuable roadmap for future
studies.
**Acknowledgement:**
Z. Zhang acknowledges the financial support from the Regional and Global Climate Modeling
Program (Grant DE-SC0014641) funded by the Office of Biological and Environmental Research
in the US DOE Office of Science. V. Larson is grateful for financial support from Climate Model
Development and Validation grant DE-SC0016287, which is funded by the Office of Biological
and Environmental Research in the US DOE Office of Science. M. Wang was supported by the
Minister of Science and Technology of China (2017YFA0604001).The computations in this study
were performed at the UMBC High Performance Computing Facility (HPCF). The facility is
supported by the U.S. National Science Foundation through the MRI program (Grants CNS-
0821258 and CNS-1228778) and the SCREMS program (Grant DMS-0821311), with substantial
support from UMBC.






Figures:

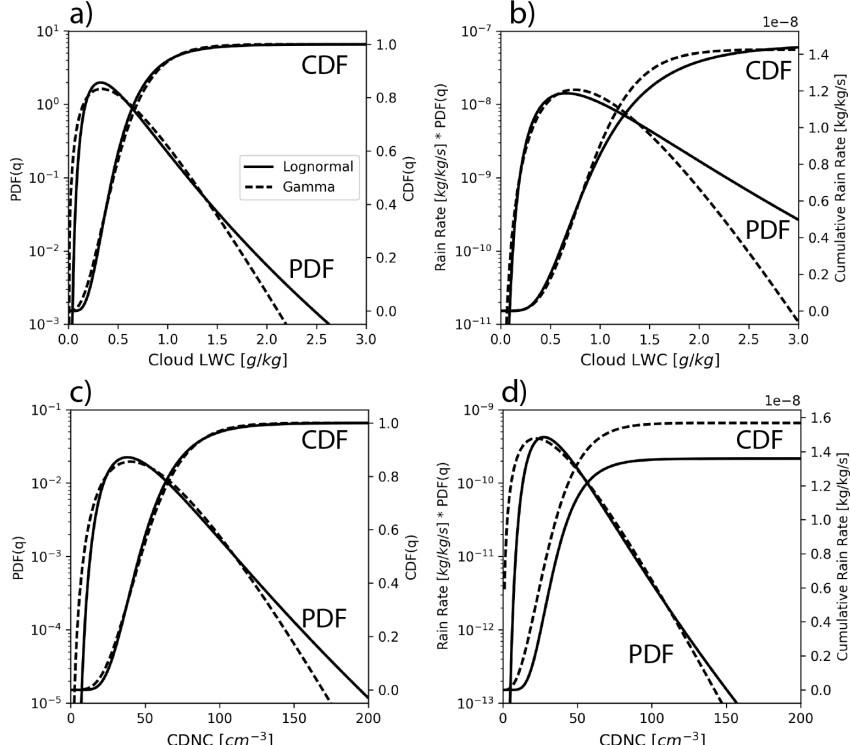


*Figure 1* a) The PDF and cumulative distribution function (CDF) of cloud LWC that follow the
Gamma (dashed) and Lognormal (solid) distribution. For the both distributions, $\langle LWC \rangle =$
$0.5 g/kg$ and $v = 3.0$. b) The PDF and CDF of rain rate computed based on the KK2000 scheme
in Eq. (12) and the PDF of LWC. In the computation, the CDNC is kept at a constant of 50 $cm^{-1}$.
c) The PDF and CDF of CDNC that follow the Gamma (dashed) and Lognormal (solid)
distribution. For the both distributions, $\langle N_c \rangle = 50 cm^{-3}$ and $v = 5.0$. d) the PDF and CDF of the
rain rate computed based on the KK2000 scheme in Eq. (12) and the PDF of CDNC. The LWC is
kept at $0.5 g/kg$ in the computation.





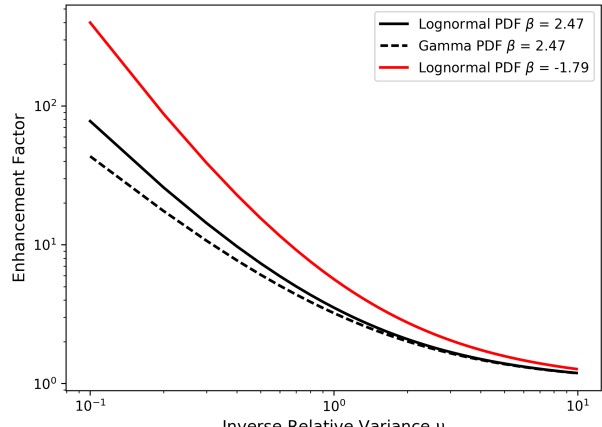


*Figure 2 Enhancement factors based on Lognormal $E(P_L, \beta)$ and Gamma $E(P_G, \beta)$ subgrid PDF*
*for different $\beta$ as a function of the inverse relative variance $\nu$.*




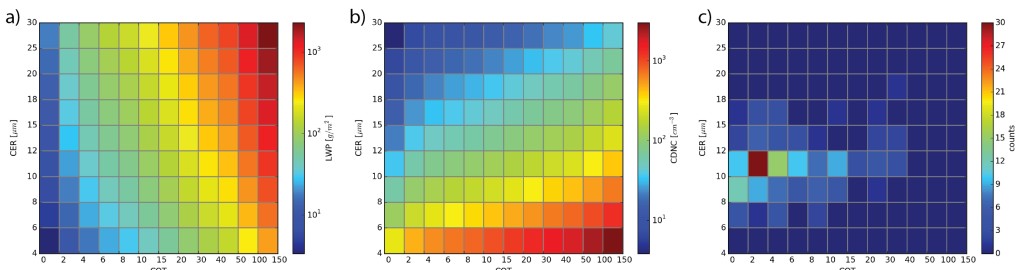

*Figure 3 The (a) LWP and (b) CDNC as a function of COT and CER. (c) An exmaple of the COT-CER*
*joint histogram observed by Aqua-MODIS on Jan. 09$^{th}$, 2007 at 1°S and 1°W.*






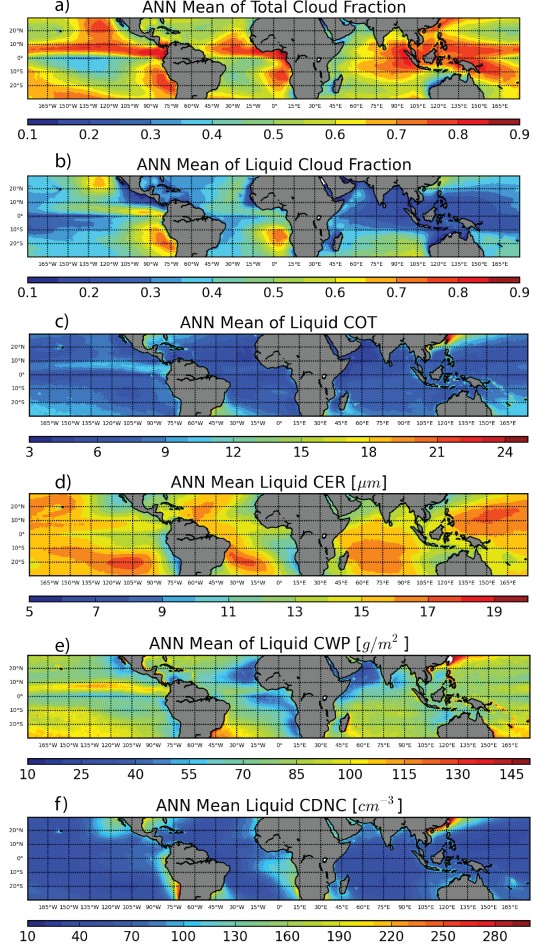


*Figure 4* 10-year (2007~2016) averaged annual mean a) total cloud fraction, b) liquid cloud
fraction, c) cloud optical thickness, d) cloud effective radius retrieved from the 3.7 μm band, e)
cloud wather path and f) cloud droplet concentration retrievals from Aqua-MODIS over the
tropical (30° S-30° N) oceans. All quantaties are "in-cloud" mean that are averaged over the
cloudy-part of the grid only.






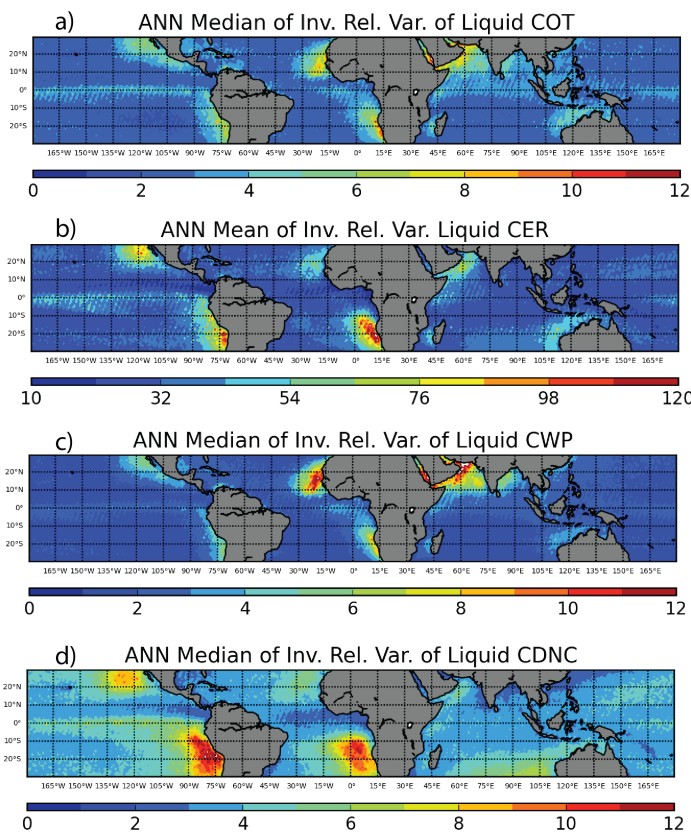


*Figure 5* 10-year (2007~2016) averaged annual mean inverse relative variance (i.e., $v = \langle x \rangle^2 / Var(x)$) of a) COT, b) CER, c) LWP and d) CDNC. Note that the color scale of CER is different from others'.







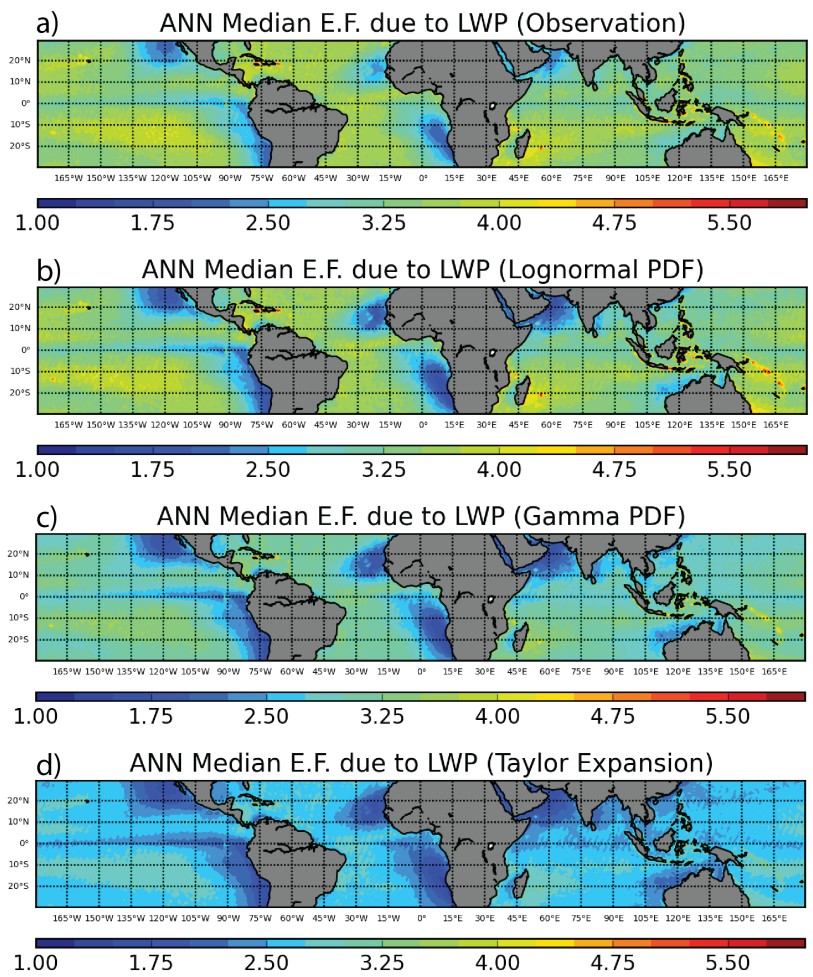

*Figure 6* The median enhancement factor for the KK2000 scheme due to subgrid variation of
LWP computed a) directly from observation, i.e., $E_q$ in Eq. (17), b) from relative variance
assuming Lognormal PDF of LWP and c) from relative variance assuming the Gamma PDF of
LWP.







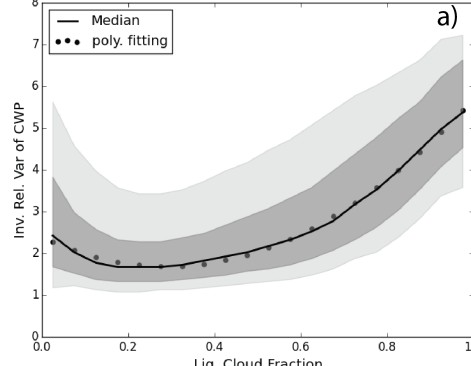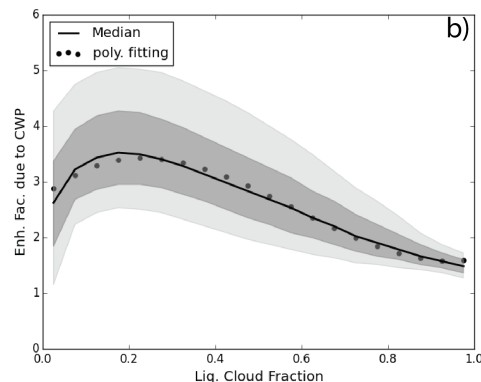


*Figure 7* a) The inverse relative variance $v$ and b) autoconversion enhancement factor due to
LWP subgrid variability assuming Log-normal PDF as a function of grid-mean liquid cloud
fraction, where the solid line, dark shaded area, and light shaded area correspond to the
median value, 25%~75% percentiles, and 10~90% percentiles, respectively. The dotted lines
correspond to simple 3-rd order polynomial fitting.





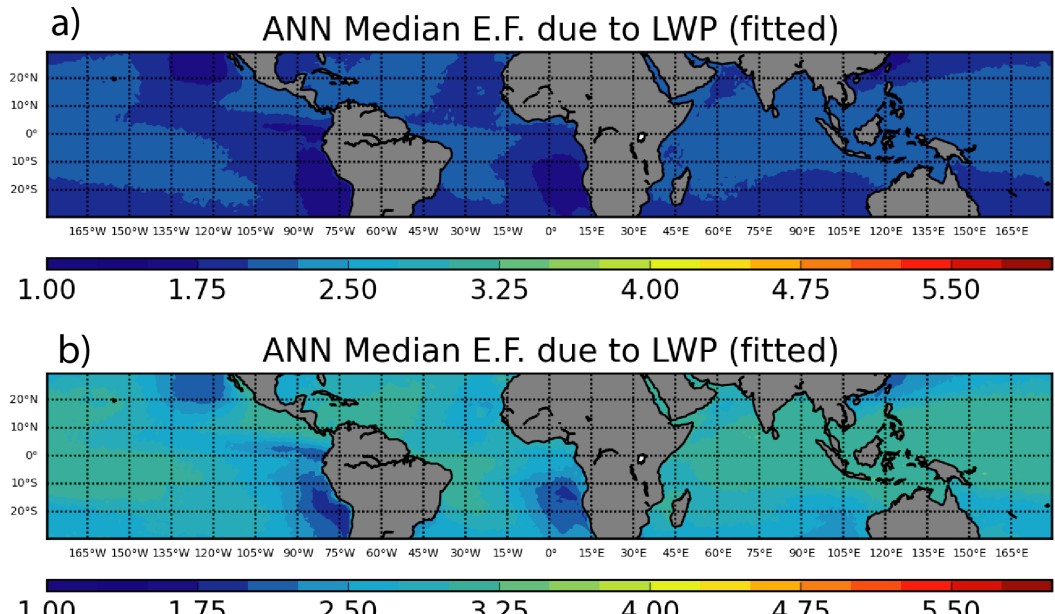

*Figure 8 Median value of the enhancement factor computed based on the a) $v(f_{liq})$*
*parameterization scheme in Eq. (26) and b) $E_q(f_{liq})$ parameterization scheme in Eq. (27).*






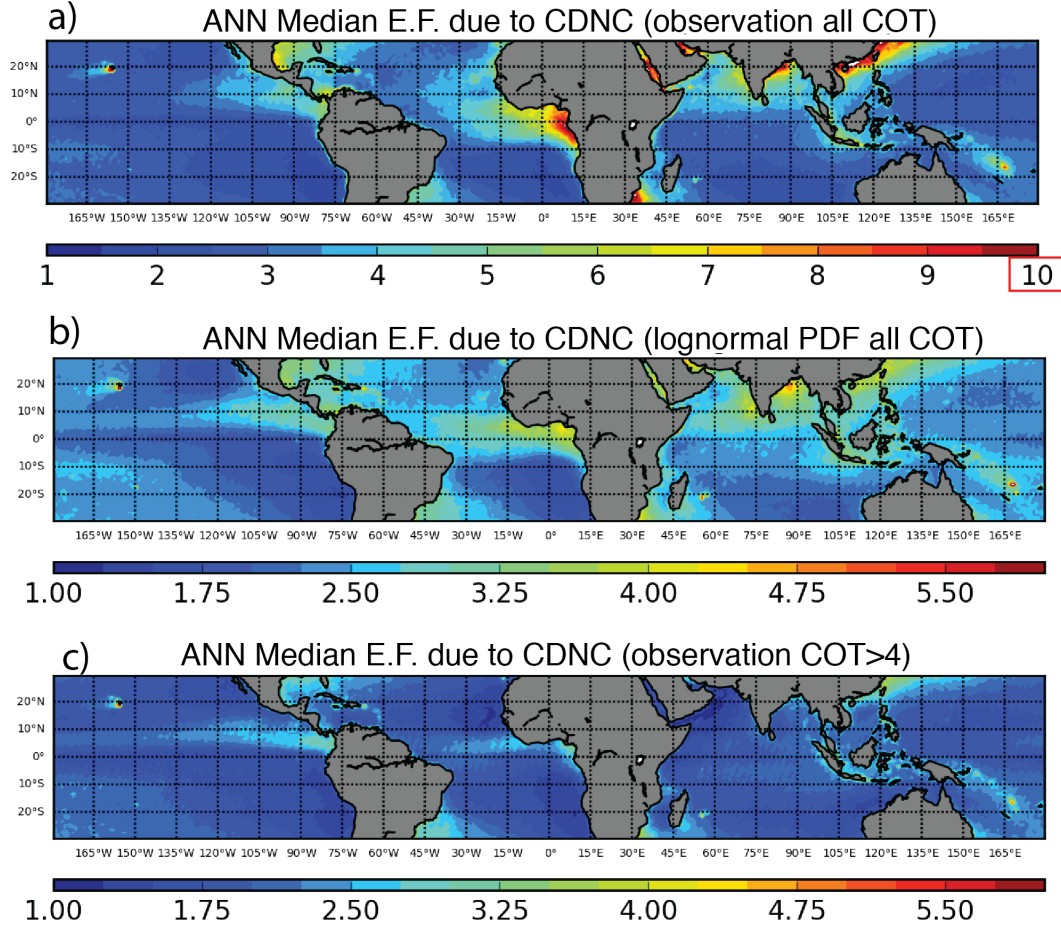


*Figure 9 Median value of the enhancement factor $E_N$ derived from a) observation based on Eq.*
*(19) and b) from Eq. (21) assuming Lognormal subgrid CDNC distribution. c) same as a) except*
*that thin clouds with COT <4 have been screened out from the analysis.*




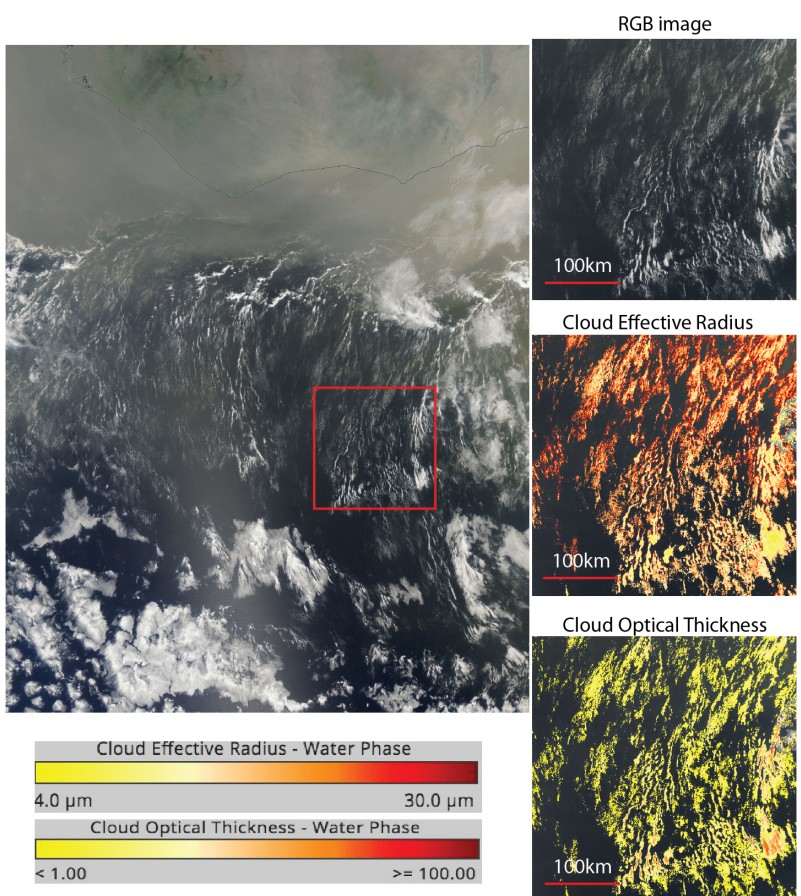

*Figure 10 An example of the large $E_N$ in the Gulf of Guinea observed by Aqua-MODIS on*
*Jan.09$^{th}$, 2007. The large image on the left shows the true color image of the region. The three*
*smaller images on the right are, from top to bottom, the zoom-in RGB image, CER and COT*
*retrievals of the subregion in red box.*





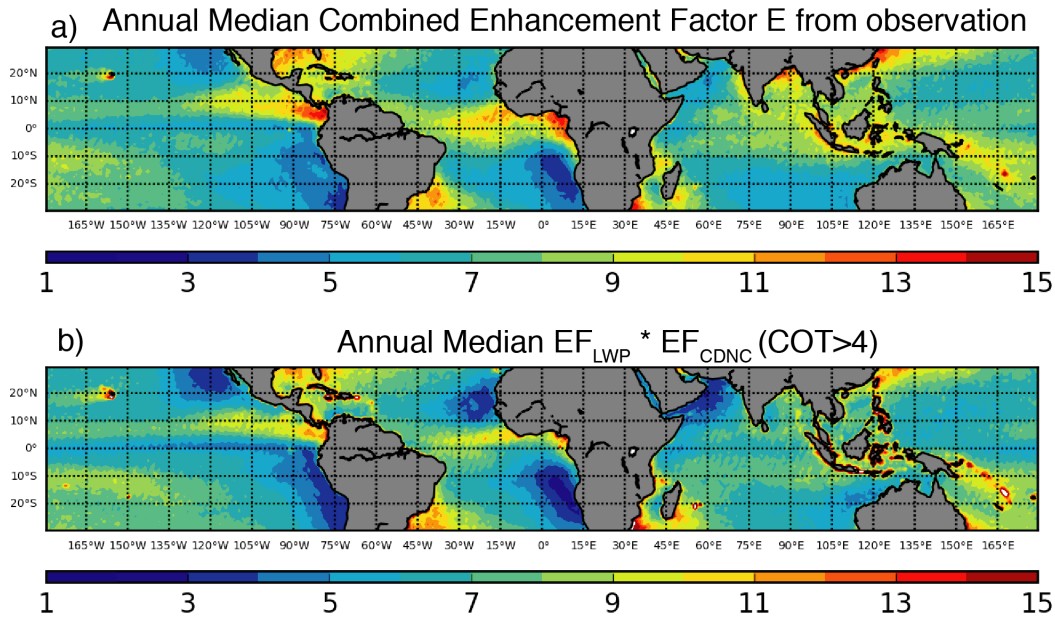

*Figure 11 a) the combined enhancement factor based on Eq. (15), b) the combined enhancement factor based on the assumption that subgrid variations of LWP and CDNC are uncorrelated, i.e., $E_q \cdot E_N (COT > 4)$. Optical thin clouds (COT<4) are screened out in the computation of $E_N$ to reduce the impact of retrieval artifacts.*



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
