# Peer review of "Subgrid Variations of the Cloud Water and Droplet Number"

_Atmospheric Chemistry and Physics, 2018_

## Referee Comment (RC1) · M. Lebsock (Referee) · 28 Aug 2018

General comments:

This paper uses Daily gridded Level-3 histograms of MODIS cloud retrievals to derive the small-scale variability in liquid cloud properties, specifically cloud liquid water path (lwp) and droplet number concentration(cdnc). This is the first study to address the variance of cdnc from satellite data. The variability is then used to diagnose the expected enhancement of the autoconversion process due to sub-grid scale distribution

of cloud fields in global models. The regional variation of the enhancement are shown. Surprisingly the enhancement due to variability in the cdnc is shown to be often larger than that due to lwp.

The largest enhancement due to number concentration variability is correlated with number concentration itself. This correlation is largely unexplained and a major result of the paper. There is a limited attempt to attempt to explain the unexpectedly large cdnc enhancement factor based on retrieval uncertainty in broken cloud scenes but the authors should consider physical mechanisms as well. I would suggest that thin detrained veil clouds near precipitating cumulus could be a physical mechanism for seeing this variability in the observations.

The science focus of this paper is novel and timely, the methodology is appropriate, and the presentation is generally good. I've included some additional references to add and specific comments below. In terms of additional analysis I would advocate quantifying the correlation between E_n and other cloud properties on various scales (correlate 1 degree grids (super pixel), correlate spatial patterns) to identify the controlling factors. This will help us better understand what variables might be influencing the high E_n (i.e. cloud fraction, low optical depth, CDNC, LWP, etc. A Table might work well to present these results.

-Matt Lebsock

Specific comments:

Lines 123-128: add Ahlgrimm et al., 2016 (https://doi.org/10.1002/qj.2783). They also use DOE data and create a parameterization of E based on cloud fraction.

Line 135: Add citation to Takahashi et al., 2017 (https://doi.org/10.1002/2016JD026404 ). They have shown that more advance parameterization, specifically a version of the Multi-scale Modeling Framework model is able to produce reasonable distributions of regional distributions of the cloud water heterogeneity when compared against the

satellite observations (their figure 2).

Line 137: Somewhere in here you should point out that the estimate of variance depends on the spatial resolution of the observations. With satellite observations (even MODIS) we are using relatively coarse observations and therefore we cannot resolve variance on the smallest scales. So satellite observations will necessarily underestimate variance because of this effect, however, they should provide an accurate assessment of regional distributions of the microphysical process enhancement factors.

Line 146: I wouldn't say that the 'empty cloud' problem is a well defined term. I can guess what this means but I would state explicitly a diagnosis of the problem. Probably there is too much rain and clouds with very low or zero liquid water path at the end of the time step???

Line 246: I think that E_q should be E_N here and cloud water should be CDNC.

Figure 1B/Line 262/Line263, and elsewhere: What is plotted here is not the rain rate. It is rate of conversion cloud water to precipitation water (or the autoconversion process rate). Rain rate is the integral of over the precipitation drop size distribution multiplied by the density-dependent fall velocity for each drop radius. This should be corrected throughout the manuscript.

Line 318: You should point out that calculating the nu parameter in this way can be very sensitive to outliers as the sample size gets small (i.e. low cloud fraction) and there are other methods to calculate nu from the data (e.g. Oreopoulos and Cahalan, 2005) that will give different answers.

Figure 5: The caption says these are means, as does panel b. But the other panels say median as does the paper text. Which is it? Median I think....

Line 327: Lebsock et al., 2011 (https://doi.org/10.1175/2010JAMC2494.1) also argue this about 3.7 micron re.

Figure 8: you should describe in the caption the difference in the fitting so the reader

doesn't have to go to the text and equations.

Figure 8: I do think it is useful to show that the parameterization of nu based on cloud fraction does not work because of the non-linearity in the process. However, can you explain why even the parameterization of the enhancement factor directly on cloud fraction under-predicts the direct calculation? That relationship show in 7b is fairly normally distributed so I can't understand why the parameterization would not get the median about right.

Eqs. 26/27 and related discussion: I don't like this parameterization of nu based on cloud fraction because it isn't well justified physically. Ideally both the cloud fraction and nu could be calculated from either prognostic or diagnostic distribution of the subgrid co-variability of total water and temperature. CLUBB in fact can do this so there should be no need to for such an ad-hoc representation. It is true that such relationships have been advocated in the past but they strike me as very unphysical. I wouldn't advocate this in the context of CLUBB, which is heavily referenced here.

Line 603: One physical interpretation of the MODIS retrievals of high effective radii in these broken cloud scenes is that they could be 'optically thin veil' clouds as described by O et al. (https://doi.org/10.1029/2018GL077084) to be extensive detrained anvil cloud from shallow cumulus with low liquid water content and very low CDNC -> thus potentially large radius. Indeed they are often seen by cloud radar (Wood et al., 2018). Now if this is the case in reality these clouds might contribute quite a bit to the variance in CDNC but shouldn't lead to any substantial increase in the autoconversion because the low CDNC pixels should also have very low liquid water path -> so the correlation should matter. In fact you show this exact correlation later on.

Line 603: I think it is important to explore and show some correlations between the E_N and various other parameters, such as CDNC, cloud fraction, number of pixels with cloud optical depth < 4. Clearly if some factor is influencing E_N (Like cloud fraction) you would expect to see some correlation between the variables. You could
show either the regional correlations, or do this for individual 1 degree grids. It seems quite clear that there is not a good correlation between liquid cloud fraction and E_N which doesn't support the idea that cloud-fraction related retrieval artifacts have much to do with these results. If on the other hand the E_N mostly correlates with large CDNC, which I suspect it does, then there is a mystery yet to be explained.

Line 646: Again, I think that there may be a physical explanation for this correlation. Specifically that there are a lot of these low water, low N veil clouds around shallow convection.

Line 665: I would argue significantly better.

Line 683: Another example of parameterization that includes subgrid information is the EDMF approach (e.g. Sušelj et al., 2013, https://doi.org/10.1175/JAS-D-12-0106.1), variants of which are used in a number of models.

Technical comments:

Line 41 The phrasing 'clear cloud' might be confusing. Consider 'obvious' or 'demonstrable' instead of clear.

Line 94: superfluous 'on'

Line 369: the 2∼4 notation seems odd to me. I would use ∼2-4 COT and ∼10-12 um.

Line 402: 'dominate' -> 'dominant'

Line 458: missing 'of'

Line 480: grammar, missing word after more.

Line 493: 'product' -> 'production'

Line 496: second 6b should be 6a.

Line 497: 'tend' -> 'tends'

Line 508: 'facts' -> 'fact'

Eqs. 26/27: parenthesis don't match.

---

## Referee Comment (RC2) · Anonymous Referee #1 · 28 Aug 2018

Authors derived the subgrid variations of liquid-phase cloud properties over the tropical ocean and investigated the autoconversion enhancement factors using MODIS product. This paper is well written, and of relevance to a broad audience. It is worthy of publication subject to the following issue.

(1) Authors assumed that subgrid variation of LWC could be inferred from the spatial variability of LWP. LWP is the vertical integrated LWC over cloud depth, so its subgrid variations include cloud depth variations. But LWC's variations does not. Please justify this assumption.

[Figure]

Typos: (1) Lines 359 "...any type of data quality-based data ", Should be "...any type of quality-based data".

(2) Lines 396 "...On the hand ", Should be "...On the other hand".

(3) Lines 466-467 "...Figure 6 b derived directly from the observation", Should be "...Figure 6 a derived directly from the observation."

---

## Referee Comment (RC3) · Anonymous Referee #3 · 5 Sep 2018

This paper discusses the GCM sub-grid scale variability of cloud water content and droplet number observed by MODIS, and the consequences this variability has for autoconversion parametrization in GCMs. This has become a popular topic in recent years with many papers discussing the cloud water content variability, although the attempts to discuss droplet number variability are particularly novel and welcome in this study. The paper is well written and interesting. I have compiled a list of relatively minor comments or suggestions that the authors may wish to consider.

General comment - the paper is very long, I'd encourage the authors to look for oppor-

tunities to be more concise in their descriptions and refrain from repetition of points.

L57 - the reference here should be Boutle et al. (2014, QJ) not Boutle & Abel (2012)

L60-62 - would be good to clarify a couple of things in these lines. Firstly, I think it would be better to refer to autoconversion and accretion "parametrizations" rather than "processes" - we shouldn't confuse the way we parametrize these things with physical reality, as there is not much overlap! Secondly, you should also clarify that you are ignoring variability in rain water content (qr) or Nc, as the nonlinearity of these (and correlations with qc) could strongly influence the result.

L92-95, 99-106 - this reads a little harshly on Boutle et al. (2014), who also used CloudSat data in their analysis to give a global perspective (and discussed the increase in variability from Sc to Cu and importance of co-variability on accretion). It might be worth mentioning the study of Hill et al. (2015, QJ) here as well, who extended this work to explicitly build in the regime dependence to the parametrizations. Also there is a typo on L104/5, which should say "cloud water variance is larger over the Cu region than over the Sc region".

L117-118 - again, might be good to clarify here - Boutle et al. (2014) and Lebsock et al. (2013) discuss the variation in rain water (which is distinct from cloud water). But you are correct that I'm also unaware of any studies looking at CDNC variability.

L194 - would be good to clarify here - it's not the LES that was important in KK2000, but the fact that they used a bin-resolved microphysics scheme, which accurately represented the physical processes of collision-coalescence, to derive the simple parametrizations

Figure 1d - I cannot see this referred to at all in the text, yet it shows something interesting/puzzling to me, namely a significantly different CDF of rain rate for the gamma and lognormal distributions of CDNC - can you explain why this is?

L256-260 - I'd always thought part of the argument for ignoring Nc is that its value

is typically linked to the underlying aerosol distribution, which varies on much larger spatial scales than qc, therefore the amount of Nc variability that would be 'sub-grid' is expected to be small/negligible.

L274 - please define CER as this has not been previously defined

L278 - brackets should be around the years only of Platnick et al. (2013,2017)

L344 - I'd say the current generation of GCMs are those being used for CMIP6, so perhaps update this and the reference (although 1x1degree still doesn't seem unreasonable for what many models are running)

L372 - should say "dominant" cloud types

L450 - I think there is something missing from this sentence - "this approach is more although it may be..."

Figure 6d - does not appear to be referred to in the text. It should either be discussed why it is relevant, or not shown.

L487-501 - given there is already a parametrization of v(f_liq) in existence, namely that of Boutle et al. (2014), it would be interesting and very easy to see how well their parametrization compares to the independent MODIS dataset generated in this study. It's probably too much work to investigate the Hill et al (2015) parametrization as that would require a way of determining from MODIS whether cloud is convective or not, but that would also be interesting.

L506-516 - it would also be worth noting that these fits are only applicable to a single model resolution, and so not as useful as existing parametrizations with inbuilt scale-adaptiveness.

L559 - there is reference to supplementary materials, yet I cannot find any?
* * *
[Figure]

2018.

---

## Author Response (AR1)

Response to Reviewers

Response to Reviewer #1 Matthew Lebsock

I would like to thank Dr. Matthew Lebsock for his insightful and suggestive comments that helped us substantially improve the manuscript. Point-to-point replies to the comments are provided below (reviewer's comments in italic blue font).

*General comments:*

*This paper uses Daily gridded Level-3 histograms of MODIS cloud retrievals to derive the small-scale variability in liquid cloud properties, specifically cloud liquid water path (lwp) and droplet number concentration(cdnc). This is the first study to address the variance of cdnc from satellite data. The variability is then used to diagnose the expected enhancement of the autoconversion process due to sub-grid scale distribution of cloud fields in global models. The regional variation of the enhancement are shown. Surprisingly the enhancement due to variability in the cdnc is shown to be often larger than that due to lwp.*

*The largest enhancement due to number concentration variability is correlated with number concentration itself. This correlation is largely unexplained and a major result of the paper. There is a limited attempt to attempt to explain the unexpectedly large cdnc enhancement factor based on retrieval uncertainty in broken cloud scenes but the authors should consider physical mechanisms as well. I would suggest that thin detrained veil clouds near precipitating cumulus could be a physical mechanism for seeing this variability in the observations.*

*The science focus of this paper is novel and timely, the methodology is appropriate, and the presentation is generally good. I've included some additional references to add and specific comments below. In terms of additional analysis I would advocate quantifying the correlation between E_n and other cloud properties on various scales (correlate 1 degree grids (super pixel), correlate spatial patterns) to identify the controlling factors. This will help us better understand what variables might be influencing the high E_n (i.e. cloud fraction, low optical depth, CDNC, LWP, etc. A Table might work well to present these results.*

Reply: Thanks for the review and helpful comments. Following your suggestions, we made significant revisions to the paper. Major changes include:

- We added more discussions on the correlation between LWP and CDNC and its implications for enhancement factor.
- We also provide some possible physical explanation on the large $E_N$. Please see details below.
- Figure 5, 7, 10, 11 are updated.

*Specific comments:*

*Lines 123-128: add Ahlgrimm et al., 2016 (https://doi.org/10.1002/qj.2783). They also use DOE data and create a parameterization of E based on cloud fraction.*
Reply: Thanks. The paper is added to the citation list.

*Line 135: Add citation to Takahashi et al., 2017 (https://doi.org/10.1002/2016JD026404). They have shown that more advance parameterization, specifically a version of the Multi-scale Modeling Framework model is able to produce reasonable distributions of regional distributions of the cloud water heterogeneity when compared against the satellite observations (their figure 2).*
Reply: Thanks. The paper is added to the citation list.

*Line 137: Somewhere in here you should point out that the estimate of variance de-pends on the spatial resolution of the observations. With satellite observations (even MODIS) we are using relatively coarse observations and therefore we cannot resolve variance on the smallest scales. So satellite observations will necessarily underestimate variance because of this effect, however, they should provide an accurate assessment of regional distributions of the microphysical process enhancement factors.*
Reply: Good point. Some discussions on the limitations of satellite observations are added after the Lebsock (2013) study.

*Line 146: I wouldn't say that the 'empty cloud' problem is a well defined term. I can guess what this means but I would state explicitly a diagnosis of the problem. Probably there is too much rain and clouds with very low or zero liquid water path at the end of the time step?*
Reply: You are right. "empty clouds" have near-zero cloud water which is caused by excessive rain (Song et al. 2018). This sentence is revised.

*Line 246: I think that $E\_q$ should be $E\_N$ here and cloud water should be CDNC.*

Reply: Thanks for catching this. It is revised.

*Figure 1B/Line 262/Line263, and elsewhere: What is plotted here is not the rain rate. It is rate of conversion cloud water to precipitation water (or the autoconversion process rate). Rain rate is the integral of over the precipitation drop size distribution multiplied by the density-dependent fall velocity for each drop radius. This should be corrected throughout the manuscript.*

Reply: Thanks for pointing this out. We should be more careful. In the revised manuscript, we use "autoconversion rate", instead of "rain rate" throughout the paper.

*Line 318: You should point out that calculating the nu parameter in this way can be very sensitive to outliers as the sample size gets small (i.e. low cloud fraction) and there are other methods to calculate nu from the data (e.g. Oreopoulos and Cahalan, 2005) that will give different answers.*

Reply: Thanks for pointing this out. Indeed, the method we used in this study is the method of moment (MOM). The inverse relative variance can also be estimated using the maximum likelihood estimate (MLE). We pointed this out in the revised manuscript.

The MODIS level 3 product reports the logarithm mean of cloud optical thickness which enables us to use the MLE method to estimate the $\nu_{MLE}$ from Eq. 6 of Oreopoulos and Cahalan (2005). The results are shown below compared with the value from the MOM $\nu_{MOM}$. Apparently, $\nu_{MLE}$ tends to be larger than $\nu_{MOM}$ especially over regions with low water cloud fraction, although the spatial pattern is similar. This is probably because, as you pointed out, the MOM is more prone to the impact of extreme values when cloud fraction is small. Nevertheless, the difference does not change any conclusions.

[Figure]

*Figure 5: The caption says these are means, as does panel b. But the other panels say median as does the paper text. Which is it? Median I think. . ..*

Reply: It's a typo and should be "Median". Corrected.

*Line 327: Lebsock et al., 2011 (https://doi.org/10.1175/2010JAMC2494.1) also argue this about 3.7 micron re.*

Reply: Thanks. This paper is cited in the revised version. Of course, the choice of coefficient for LWP computation does not matter in this study because it is a common factor in both numerator and denominator in the calculation of $v$.

*Figure 8: you should describe in the caption the difference in the fitting so the reader doesn't have to go to the text and equations.*

Reply: Good suggestion. We added the information.

*Figure 8: I do think it is useful to show that the parameterization of nu based on cloud fraction does not work because of the non-linearity in the process. However, can you explain why even the parameterization of the enhancement factor directly on cloud fraction under-predicts the direct calculation? That relationship show in 7b is fairly normally distributed so I can't understand why the parameterization would not get the median about right.*

Reply: This is a very good question. To answer it, first let us explain how $E_q$ in Figure 6 and Fig. 8 are obtained. The parameterization scheme in Eq. (27) and Fig. 7 are developed based on the relation between monthly-mean observation-based $E_q$ and monthly-mean $f_{liq}$ in the tropics (i.e., 10 years x 12months x 360 longitude x 60 latitude x fraction of ocean). The sample size would be too large if daily products were used. After we obtained the parameterization scheme (i.e., Eq. 27), we then used it to compute the daily $E_q$ based on daily $CF_{liq}$. The daily $E_q$ values are then temporally aggregated, weighted by daily $f_{liq}$, to first obtain monthly and then annual $E_q$ in Fig. 8b in the same way as we obtain observed $E_q$ in Fig. 6. Going back to your question, we think the underestimation of parameterized $E_q$ (Figure 8b compared to Figure 6a) is due to the fact that the parameterization is developed based on monthly data but applied to daily $f_{liq}$. To test this, we applied the parameterization scheme to monthly $f_{liq}$. The results are significantly better. See below.

[Figure]

a) ANN Median E.F. due to LWP (Observation)

b) ANN Median E.F. due to LWP (Daily CF)

c) ANN Median E.F. due to LWP (Monthly CF)

The lesson learned is that the simple parameterization scheme developed based on monthly $f_{liq}$ cannot capture the day-to-day variation of $E_q$, which is not surprising. In our view, the parameterization scheme is only better than assuming a constant $E_q$ in the sense that it can capture the cloud regime dependence. However, it would be unrealistic to hope that it can simulate the dramatic instantaneous variation. For that, we would have to rely on advanced scheme like CLUBB or MMF.

*Eqs. 26/27 and related discussion: I don't like this parameterization of nu based on cloud fraction because it isn't well justified physically. Ideally both the cloud fraction and nu could be calculated from either prognostic or diagnostic distribution of the subgrid co-variability of total water and temperature. CLUBB in fact can do this so there should be no need to for such an ad-hoc representation. It is true that such relationships have been advocated in the past but they strike me as very unphysical. I wouldn't advocate this in the context of CLUBB, which is heavily referenced here.*

Reply: We agree with your point about the parameterization of $\nu$. The highly non-linear relation between $\nu$ and the enhancement factor makes the parameterization not so useful. It is shown here simply because some previous studies, e.g. Boulte et al. (2014), Xie and Zhang (2015), had tried to parameterize the $\nu$ directly. The unsatisfying results motived us to parameterize the enhancement factor directly.

On the other hand, we think the direct parameterization of enhancement factor is meaningful. It provides with a simple way for those GCMs without advanced sub-grid parameterization scheme to account for the impacts of cloud inhomogeneity on precipitation simulation. We agree that CLUBB presumably would do a better job than simple parameterization. Nevertheless, the results from this study, including the parameterization of enhancement factor, provide observational basis for evaluating the results from CLUBB.

*Line 603: One physical interpretation of the MODIS retrievals of high effective radii in these broken cloud scenes is that they could be 'optically thin veil' clouds as described by O et al. (https://doi.org/10.1029/2018GL077084) to be extensive detrained anvil cloud from shallow cumulus with low liquid water content and very low CDNC -> thus potentially large radius. Indeed they are often seen by cloud radar (Wood et al., 2018). Now if this is the case in reality these clouds might contribute quite a bit to the variance in CDNC but shouldn't lead to any substantial increase in the autoconversion because the low CDNC pixels should also have very low liquid water path -> so the correlation should matter. In fact you show this exact correlation later on.*

Reply: This is a very insightful comment and thanks for the references (we are aware of Wood et al. 2018 but not O et al.). Following your suggestions, we have de-emphasized the influence of retrieval error and focused more on the potential physical processes that lead to the large subgrid CDNC variance. These papers are now in Section 4 when discuss the new Figure 5 e about the correlation between LWP and CDNC and the its implications.

*Line 603: I think it is important to explore and show some correlations between the E_N and various other parameters, such as CDNC, cloud fraction, number of pixels with cloud optical depth < 4. Clearly if some factor is influencing E_N (Like cloud fraction) you would expect to see some correlation between the variables. You could show either the regional correlations, or do this for individual 1 degree grids. It seems quite clear that there is not a good correlation between liquid cloud fraction and E_N which doesn't support the idea that cloud-fraction related retrieval artifacts have much to do with these results.  If on the other hand the E_N mostly correlates with large CDNC, which I suspect it does, then there is a mystery yet to be explained.*

Reply: Thanks for the great suggestions! We made several significant changes to the paper accordingly. We replaced original Figure 10 (which focuses on the retrieval artifacts) with an analysis of the dependence of $E_N$ on liquid cloud fraction and CDNC. See below. As you suspected, $E_N$ shows a stronger dependence on CDNC than cloud fraction, which seems to suggest that the dependence is largely due to some underlying physical mechanisms rather than retrieval artifacts. The largest $E_N$ is usually found where CDNC is large and cloud fraction is small and it decreases with decreasing CDNC and to a less extent also with increasing cloud fraction. The strong dependence of $E_N$ on CDNC might be explained by the following mechanism in which aerosol plays an important role: when aerosol loading is small, even weak updraft can activate most CCN. As a result, the subgrid turbulence and variance of thermodynamical conditions are not importance leading to small $E_N$. In contrast, when aerosol loading is large, subgrid variations of updraft and thermodynamical conditions could lead to significant subgrid variations of CDNC, leading to large $E_N$.

In addition to the analysis $E_N$, we also added some more in-depth explanation of the importance of LWP and CDNC correlation on enhancement factor simulation at the end of Section 2.2. First, a formula for *combined* enhancement based on the bi-variate lognormal distribution is presented (Eq. 22). Second, we pointed out that the current GCMS, even those with advanced sub-grid parameterization such as CLUBB, only consider the enhancement factor due to LWP $E_q$, the effect of $E_N$ and the correlation term $E_{COV}$ are ignored. Moreover, an equation is added (Eq. 25) to explain under what circumstances would $E_q$ underestimate or overestimate the combined effect $E_q \cdot E_N \cdot E_{COV}$. In addition, Figure 5 e is added to show the subgrid correlation coefficient of LWP and CDNC and in Figure 11 we discussed the importance of considering $E_{COV}$ in computing the combined enhancement factor.

We feel that these revisions, based on your suggestions, had made the paper more insightful and more revealing.

[Figure]

*Dependence of $E_N$ on $f_{liq}$ and $N_d$. The color map corresponds to the mean value of $E_N$ for a given $N_d$ and $f_{liq}$ bin. The white contour lines correspond to the relative sampling frequency of $N_d$ and $f_{liq}$ bins (i.e., the most frequently observed combination is $N_d \sim 50 cm^{-3}$ and $f_{liq} \sim 0.1$ ).*

*Line 646: Again, I think that there may be a physical explanation for this correlation. Specifically that there are a lot of these low water, low N veil clouds around shallow convection.*

Reply: See our reply above.

*Line 665: I would argue significantly better.*

Reply: agree and revised.

*Line 683: Another example of parameterization that includes subgrid information is the EDMF approach (e.g. Sušelj et al., 2013, https://doi.org/10.1175/JAS-D-12-0106.1), variants of which are used in a number of models.*

Reply: we added the EDMF as another example of "advanced subgrid cloud parameterization scheme". Thanks for pointing it out.

*Technical comments:*

*Line 41 The phrasing 'clear cloud' might be confusing. Consider 'obvious' or 'demonstrable' instead of clear.*

*Line 94: superfluous 'on'*

*Line 369: the 2~4 notation seems odd to me. I would use ~2-4 COT and ~10-12 um.*

*Line 402: 'dominate' -> 'dominant'*

*Line 458: missing 'of'*

*Line 480: grammar, missing word after more. Line*

*493: 'product' -> 'production'*

*Line 496: second 6b should be 6a. Line 497: 'tend' -> 'tends'*

*Line 508: 'facts' -> 'fact'*

*Eqs. 26/27: parenthesis don't match.*

Reply: Thanks a lot for catching these typos and mistakes. They are all corrected.

I would like to thank the reviewer for the comments and suggestions. Point-to-point replies to the comments are provided below (reviewer's comments in italic blue font).

*Authors derived the subgrid variations of liquid-phase cloud properties over the tropical ocean and investigated the autoconversion enhancement factors using MODIS product. This paper is well written, and of relevance to a broad audience. It is worthy of publication subject to the following issue.*

*(1) Authors assumed that subgrid variation of LWC could be inferred from the spatial variability of LWP. LWP is the vertical integrated LWC over cloud depth, so its subgrid variations include cloud depth variations. But LWC's variations does not. Please justify this assumption.*

Reply: Indeed, MODIS retrievals only provide the LWP instead of the vertically resolved LWC retrieval. This is an important limitation of this study which we pointed out clearly in Section 3.

However, as we also pointed out, other techniques face more or less similar challenge. "We note here that it is the LWC $q_c$, instead of the LWP, that is used in the KK2000 scheme. So, the spatial variability of LWC is what is most relevant. However, the remote sensing of cloud water vertical profile from satellite sensor for liquid-phase clouds is extremely challenging even with active sensors. It is why most previous studies using the satellite observations analyzed the spatial variation of LWP, rather than LWC. In fact, even Lebsock et al. (2013), who used the level-2 CloudSat observations, had to use the vertical averaged LWC in their analysis. Airborne in situ measurement faces similar challenge. For example, Boutle et al. (2014) use the LWC observation along "horizontal flight tracks" to study the spatial variability of cloud water, which only samples the LWC at certain levels of MBL clouds. Ground-based observations are much better than satellite and airborne observation in this regard. Recently, Xie and Zhang (2015) analyzed the cloud water profiles retrieved using ground-based radars from the three ARM sites and found no obvious in-cloud vertical dependence of the spatial variability of LWC."

*Typos: (1) Lines 359 ": : :any type of data quality-based data ", Should be ": : :any type of quality-based data".*

*(2) Lines 396 ": : :On the hand ", Should be ": : :On the other hand".*

*(3) Lines 466-467 "...Figure 6 b derived directly from the observation", Should be ": : :Figure 6 a derived directly from the observation."*

Reply: thanks for catching these typos. They are all corrected

I would like to the reviewer for the insightful and suggestive comments that helped us substantially improve the manuscript. Point-to-point replies to the comments are provided below (reviewer's comments in italic blue font).

*This paper discusses the GCM sub-grid scale variability of cloud water content and droplet number observed by MODIS, and the consequences this variability has for autoconversion parametrization in GCMs. This has become a popular topic in recent years with many papers discussing the cloud water content variability, although the attempts to discuss droplet number variability are particularly novel and welcome in this study. The paper is well written and interesting. I have compiled a list of relatively minor comments or suggestions that the authors may wish to consider.*

*General comment –*
*the paper is very long, I'd encourage the authors to look for opportunities to be more concise in their descriptions and refrain from repetition of points.*
Reply: The theoretical background part is longer than we hoped but necessary so the readers to understand the studies that followed. The length of the revised version is reduced by one page. It is not trivial considering that we extend the scope of the research significantly.

*L57 - the reference here should be Boutle et al. (2014, QJ) not Boutle & Abel (2012)*
Reply: we updated the references.

*L60-62 - would be good to clarify a couple of things in these lines. Firstly, I think it would be better to refer to autoconversion and accretion "parametrizations" rather than "processes" - we shouldn't confuse the way we parametrize these things with physical reality, as there is not much overlap! Secondly, you should also clarify that you are ignoring variability in rain water content (qr) or Nc, as the nonlinearity of these (and correlations with qc) could strongly influence the result.*
Reply: Agree, the KK2000 is simply a parameterization based on the least-square fitting to the LES results. We change the wording from "process" to "parameterization" throughout the text whenever appropriate.

We pointed out at the beginning of section 2.2 that we will only focus on the simulation of autoconversion while other processes such as accretion have been investigated in previous studies.

*L92-95, 99-106 - this reads a little harshly on Boutle et al. (2014), who also used CloudSat data in their analysis to give a global perspective (and discussed the increase in variability from Sc to Cu and importance of co-variability on accretion). It might be worth mentioning the study of Hill et al. (2015, QJ) here as well, who extended this work to explicitly build in the regime dependence to the parametrizations. Also there is a typo on L104/5, which should say "cloud water variance is larger over the Cu region than over the Sc region".*

Reply: Agree, we revised the discussion, added the Hill et al. (2015) and also corrected the typo.

*L117-118 - again, might be good to clarify here - Boutle et al. (2014) and Lebsock et al. (2013) discuss the variation in rain water (which is distinct from cloud water). But you are correct that I'm also unaware of any studies looking at CDNC variability.*

Reply: Following your suggestion, we pointed out again that *Boutle et al. (2014) and Lebsock et al. (2013)* have investigated the variation of subgrid cloud water as well as rain water.

*L194 - would be good to clarify here - it's not the LES that was important in KK2000, but the fact that they used a bin-resolved microphysics scheme, which accurately represented the physical processes of collision-coalescence, to derive the simple parametrizations*

Reply: Agree. In the revised version, we pointed out after the introduction of KK2000 parameterization scheme that the KK2000 is "derived through a least-square fitting of the autoconversion rate results from a large-eddy simulation with bin microphysics that can simulate the process-level physics."

*Figure 1d - I cannot see this referred to at all in the text, yet it shows something interesting/puzzling to me, namely a significantly different CDF of rain rate for the gamma and lognormal distributions of CDNC - can you explain why this is?*

Reply: As shown in Figure 1, provided the same mean value and same inverse relative variance $v$, the lognormal distribution $P_L(x)$ is generally larger than the Gamma distribution $P_G(x)$. The difference is clearly visible when $x > 2.0$ in Figure 1 b. This differences in PDF gives rise to the difference in the CDF of autoconversion rate.

*L256-260 - I'd always thought part of the argument for ignoring Nc is that its value is typically linked to the underlying aerosol distribution, which varies on much larger spatial scales than qc, therefore the amount of Nc variability that would be 'sub-grid' is*

*expected to be small/negligible.*

Reply: Aerosol loading is only on part of the story. CDNC is not only determined by aerosol loading but also critically be the subgrid turbulence (i.e., updraft). Many previous studies have pointed out the importance of subgrid variations of updrafts in simulating cloud microphysics in GCM (e.g., (Morales and Nenes, 2010)). However, most GCMs lack the capability of simulating the subgrid variations of updrafts until very recently the advanced parameterization schemes, such as CLUBB and MMF, became available.

*L274 - please define CER as this has not been previously defined*

Reply: It is clarified.

*L278 - brackets should be around the years only of Platnick et al. (2013,2017)*

Reply: Corrected.

*L344 - I'd say the current generation of GCMs are those being used for CMIP6, so perhaps update this and the reference (although 1x1degree still doesn't seem unreasonable for what many models are running)*

Reply: Updated and added the new reference (Eyring et al., 2016)

*L372 - should say "dominant" cloud types*

Reply: Changed.

*L450 - I think there is something missing from this sentence - "this approach is more although it may be..."*

Reply: It should be more "efficient". Corrected.

*Figure 6d - does not appear to be referred to in the text. It should either be discussed why it is relevant, or not shown.*

Reply: It is removed. Thanks for catching this.

*L487-501 - given there is already a parametrization of v(f_liq) in existence, namely that of Boutle et al. (2014), it would be interesting and very easy to see how well their parametrization compares to the independent MODIS dataset generated in this study. It's probably too much work to investigate the Hill et al (2015) parametrization as that would require a way of determining from MODIS whether cloud is convective or not, but that would also be interesting.*

Reply: We first replicate the Figure 4 in Boutle et al. (2014) to confirm our code works consistently with the original result.

[Figure]

[Figure]

Then, we compared the parameterization scheme from *Boutle et al. (2014)* for grid size ~ 100km to out Figure 7a (the green dashed line). Apparently, there are some differences between the two especially for large cloud fraction, probably because the two studies are based on different data. Since the difference between the two studies are out of the scope of this paper. These figures are not shown in the paper.

*L506-516 - it would also be worth noting that these fits are only applicable to a single model resolution, and so not as useful as existing parametrizations with inbuilt scale adaptiveness.*

Reply: Agree and we already mentioned that this parameterization is only valid for 1x1 degree model resolution when we list the important limitation of this study.

*L559 - there is reference to supplementary materials, yet I cannot find any?*

Reply: We originally planned to add the seasonal plots (e.g., DJF and JJA) in the supplementary materials, but we found that seasonal plots do not really add any additional insights. So, we simply removed them. Sorry for the confusion.

[revised manuscript text omitted]

**Commented [V16]:** I wonder if Robin Hogan has ever compared lognormal and gamma PDFs, for ice or radar reflectivity at least.

To test the performance of this simple parameterization, we first substitute the $f_{liq}$ from MODIS daily mean level-3 product into the above equation and then use the resultant $v$ to compute the enhancement factor $E_q$. Unfortunately, the enhancement factor $E_q$ computed based on the parameterized $v(f_{liq})$ as shown in Figure 8a substantially underestimate the observation-based results in Figure 6, especially over the Cu regions. The deviation is probably because the relationship between $E_q$ and $v$ is highly nonlinear (e.g., Eq. *(8)* and *(14)*) and therefore the above parameterization scheme that only fits the = value of $v$ is not able to capture the variability of $E_q$. Based on this consideration, we tried an alternative approach. Instead of parameterization of $v$, we directly parameterize the enhancement factor $E_q$ as a function of $f_{liq}$. Figure 7b shows the variation of $E_q$ as a function of $f_{liq}$. As expected, $E_q$ generally decreases with increasing $f_{liq}$. The median value of $E_q$ is fitted with the following 3$^{rd}$ order polynomial of $f_{liq}$

$$E_q(f_{liq}) = 2.72 + 7.33 f_{liq} - 19.17 f_{liq}^2 + 10.69 f_{liq}^3, \ f_{liq} \in [0,1]. \qquad (31)$$

As shown in Figure 8b, the value of $E_q$ based on the above equation clearly agrees with the observation-based values in Figure 6 better than that based on the parameterization of $v(f_{liq})$. The elimination of the middle step indeed improves the parameterization results. While this is encouraging, it should be kept in mind that the Eq. *(31)* has very limited application, i.e., it is only useful for the autoconversion rate computation for a particular value of the autoconversion exponent beta, i.e., $\beta_q = 2.47$. A good parameterization of $v$ could be useful for not only autoconversion, but also for accretion and radiation computations. Another caution is that, if applied to a GCM, the performance of the $E_q(f_{liq})$ parameterization in Eq. *(31)* will be dependent on the simulated accuracy of $f_{liq}$ in the model.

**5.2. Influence of subgrid variance of CDNC**

Now we will investigate the impacts of subgrid CDNC variation on the autoconversion rate simulation. For the moment, we will consider $E_N$ only. The impact of CDNC and cloud water correlation will be discussed in the next section. Similar to $E_q$ we first derive $E_N$ from the CDNC PDF based on Eq. *(21)*. The annual mean result based on 10 years of MODIS observations is shown in Figure 9a. There are several intriguing points to note. First of all, the value of $E_N$ is actually

**Commented [V17]:** What value of the autoconversion exponent was assumed here?

The median value of larger than $E_q$ in Figure 9 such that we even have to use a different color scale for this plot.

Secondly, $E_N$ the regions with escalated $E_N$ seem to coincide with the downwind regions of biomass burning aerosols (e.g., Gulf of Guinea, East Coast of South Africa), air pollution (i.e.,

Eastern China Sea), and, most interestingly, active volcanos (e.g., Kilauea Hawaii and Ambae

Vanuatu). We have also checked the seasonal variation of the $E_N$ and the results also support this observation. Another interesting feature to note is that, although the dust outflow regions such as Tropical East Atlantic and Arabian Sea, have heavy aerosol loading, the value of $E_N$ there is only moderate. Figure 9b shows the value of $E_N$ computed based on Eq. (14) from the inverse relative variance of $v$, assuming that the subgrid CDNC follows a Lognormal PDF. Although the overall pattern is consistent with Figure 9a, the assumption of Lognormal PDF seems to underestimate $E_N$. A closer examination indicates that the Lognormal PDF tend to underestimate the population of clouds with small CDNC, and therefore underestimate the variance of CDNC as well as $E_N$. We did not compute the $E_N$ based on the Gamma distribution because of the singular value problem aforementioned in Section 2.1.

We could not find any previous observation-based study on the global pattern of the subgrid variation of CDNC and the corresponding $E_N$. So, it is difficult for us to corroborate our results. On one hand, the magnitude of $E_N$ is surprisingly large. As explained in Section 3, the

CDNC is estimated based on Eq. (27) from the MODIS retrieval of COT and CER. Several previous studies have shown that the sub-pixel level surface contamination, subpixel cloud inhomogeneity, and three-dimensional radiative transfer effects, can cause significant errors in the MODIS CER retrievals especially over broken cloud regions (Zhang and Platnick, 2011; Zhang et al., 2012; 2016). Given the fact that the CDNC retrieval is highly sensitive to CER error as a result of $N_d \sim r_e^{-\frac{5}{2}}$, the influence of retrieval uncertainty on subgrid CDNC variation cannot be ruled out. On the other hand, the pattern of $E_N$ in Figure 9a seems to suggest that there are some underlying physical mechanisms controlling the subgrid variation of CDNC, in which aerosols seem to play an important role. To achieve a better understanding, we analyzed the dependence of $E_N$ on liquid cloud fraction and grid-mean CDNC in Figure 10, which reveals that $E_N$ has a stronger dependence on CDNC than cloud fraction. This result seems to indicate that the pattern of $E_N$ in Figure 9 is largely determined by physical mechanisms rather than retrieval

**Commented [V18]:** Importantly (and confusingly), E_N is small in the main Sc regions off the coasts of California, Peru, and Namibia. Why aren't there variations in Nc in those regions off the coasts? Autoconversion will be not be enhanced in those regions by variations in Nc (or qc).

To me, it looks like E_N is controlled by variations in the (remote) source of aerosol, whereas E_Q is controlled by variability in cumulus clouds that is locally induced by turbulence. But neither is large in Sc.

Is it possible that the high values of E_N are an artifact of time variability in the aerosol as a plume of pollution from, e.g., a fire, meanders across the ocean? Is there such large variability in instantaneous snapshots? Can the MODIS observations work on instantaneous data instead of time averages?

**Commented [zz19R18]:** These are very good questions. I don't have clear answers at the moment. Some small scale CDNC variation is due to retrieval artifacts. But as shown later, even we screen out the COT<5 data, the results are still similar.

I'm in favor of your hypothesis about the spatial variation of E_N. In this paper, I just want to hold on the "observation" and leave the in-depth study of the causes to future work.

**Commented [V20]:** How accurate is the gamma distribution?

**Commented [V21]:** It might be reassuring to compare with aircraft observations. Do any aircraft observations show CER varying from 4 to 30 microns?

**Commented [zz22R21]:** Yes, in situ measurement would be highly useful. But it will be left for future work.

[revised manuscript text omitted]

Commented [V32]: This is within-cloud, not grid-box-averaged, right?

Commented [zz33R32]: correct

[Figure]

[Figure]

*Figure 5* Median value of the inverse relative variance (i.e., $v = \langle x \rangle^2 / Var(x)$) for a) COT, b)
CER, c) LWP and d) CDNC, and e) median value of the correlation coefficient between LWP and
CDNC derived from 10 years of MODIS observations. Note that the color scale of CER is
different from others'.

[Figure]

[Figure]

Figure 6 The annual mean factor for the KK2000 scheme due to subgrid variation of LWP computed a) directly from observation, i.e., $E_q$ in Eq.(20), b) from relative variance assuming Lognormal PDF of LWP, i.e., $E_q$ in Eq.(14) and c) from relative variance assuming the Gamma PDF of LWP, i.e., $E_q$ in Eq.(8).

[Figure]

[Figure]

*Figure 7* a) The inverse relative variance $v$ and b) autoconversion enhancement factor due to
LWP subgrid variability assuming Log-normal PDF as a function of grid-mean liquid cloud
fraction, where the solid line, dark shaded area, and light shaded area correspond to the
median value, 25%~75% percentiles, and 10~90% percentiles, respectively. The dotted lines
correspond to simple 3-rd order polynomial fitting.

[Figure]

[Figure]

*Figure 8 Annual mean value of the enhancement factor $E_N$ computed based on the a) $v(f_{liq})$ =*

*2.38−4.95$f_{liq}$ + 8.74$f_{liq}^2$ − 0.49$f_{liq}^3$ parameterization scheme in Eq. (30) and b) $E_q(f_{liq})$ =*

*2.72+7.33$f_{liq}$ − 19.17$f_{liq}^2$ + 10.69$f_{liq}^3$ parameterization scheme in Eq. (31).*

[Figure]

[Figure]

[Figure]

*Figure 9 Annual mean value of the enhancement factor $E_N$ derived from a) observation based*
*on Eq. (21) and b) from Eq. (14) assuming Lognormal subgrid CDNC distribution.*

[Figure]

$E_N$ based on observed CDNC PDF (Eq. 21)

[Figure]

*Figure 10 Dependence of $E_N$ on $f_{liq}$ and $N_d$. The color map corresponds to the mean value of $E_N$*
*for a given $N_d$ and $f_{liq}$ bin. The white contour lines correspond to the relative sampling*
*frequency of $N_d$ and $f_{liq}$ bins (i.e., the most frequently observed combination is $N_d \sim 50cm^{-3}$*
*and $f_{liq} \sim 0.1$).*

¶

¶
... [22]

[Figure]

a)    ANN mean combined $E$ based on observed joint PDF (Eq. 17)

b)    ANN mean $E_q * E_N$ without considering $E_{COV}$

c)    ANN mean combined $E$ assuming bi-variate lognormal PDF (Eq.23)

[revised manuscript text omitted]

I doubt that this is often true in nature.

| Page 12: [11] Deleted | Zhibo Zhang | 12/3/18 8:08:00 AM |
|---|---|---|

| Page 12: [12] Deleted | Zhibo Zhang | 12/3/18 8:08:00 AM |
|---|---|---|

| Page 13: [13] Deleted | Zhibo Zhang | 12/3/18 8:08:00 AM |
|---|---|---|
| Page 19: [14] Deleted | Zhibo Zhang | 12/3/18 8:08:00 AM |

1.1.

| Page 19: [15] Deleted | Zhibo Zhang | 12/3/18 8:08:00 AM |
|---|---|---|

| Page 19: [16] Deleted | Zhibo Zhang | 12/3/18 8:08:00 AM |
|---|---|---|

| Page 19: [17] Deleted | Zhibo Zhang | 12/3/18 8:08:00 AM |
| --- | --- | --- |

| Page 19: [18] Deleted | Zhibo Zhang | 12/3/18 8:08:00 AM |
| --- | --- | --- |

| Page 22: [19] Deleted | Zhibo Zhang | 12/3/18 8:08:00 AM |
| --- | --- | --- |

| Page 23: [20] Deleted | Zhibo Zhang | 12/3/18 8:08:00 AM |
| --- | --- | --- |

| Page 23: [21] Deleted | Zhibo Zhang | 12/3/18 8:08:00 AM |
| --- | --- | --- |

| Page 37: [22] Deleted | Zhibo Zhang | 12/3/18 8:08:00 AM |
| --- | --- | --- |

---

## Author Response (AR2)

I thank the authors for their modifications to the paper, which is certainly improved from the previous version. I have only one comment of any significance:

I disagree with the authors statement that comparison to Boutle et al. (2014) is out-of-scope of the current paper. Firstly, we are all in the process of model development. How is a model developer to decide which parametrization they should use if authors only ever compare their own parametrizations to the dataset used to construct the parametrization - this is no use to the community in deciding which parametrization to use, since all parametrizations will always look best when compared to the data used to derive them. But secondly, and perhaps more importantly, the comparison is directly relevant to the results of this paper. Boutle et al. (2014) is the only study that directly compares different observational estimates of the same quantity, showing how the estimate of sub-grid variability can be affected by the measurement technique used. In particular, they showed that CloudSat significantly under-estimates the true variability. The fact that the Boutle et al. (2014) data and parametrization shows more variability than the estimates presented here from MODIS suggests that MODIS could suffer from the same sampling and pixel-size issues as CloudSat (as discussed in Boutle et al. (2014)), meaning that the variability estimates provided here from MODIS could be an under-estimate of the true variability. I agree it's beyond the scope to give a full investigation of these differences, but it does need noting that these differences exist and that the MODIS variability could be an under-estimate of the true variability.

*Reply: following the suggestion, we have updated the Figure 7. In particular, we have added the parameterization in Boutle et al. (2014) to Figure 7a and added some brief discussion.*

A couple of minor comments:

L124-130 - I'd suggest moving this discussion to after L137 - it currently reads like your criticism of previous studies for lacking a global perspective applies to these papers, which is not true because they both make use of the same global CloudSat dataset as Lebsock et al. (2013) (in addition to ground based and in-situ measurements).

L451-452 - something has gone wrong with the referencing of Wood et al and O et al.

L517-534 - apologies for missing this last time, but it would be worth noting here that the somewhat bland results obtained from a simple parametrization of the mean v(f_liq) value are part of the motivation for a variable (regime-dependant) parametrization of v(f_liq), such as Hill et al. (2015).

*Reply: We have also revised the manuscript based on these minor comments.*